# Tracking life and death of carbon nitride supports in platinum-catalyzed vinyl chloride synthesis

Vera Giulimondi[1,2], Mikhail Agrachev[2,3], Sergei Kuzin [3],
José Manuel González-Acosta [4,5], Andrea Ruiz-Ferrando[4], Frank Krumeich [6],
Federica Bondino [7], Yung-Tai Chiang [1,2], Matteo Vanni [1],
Gunnar Jeschke [2,3], Núria López [4] & Javier Pérez-Ramírez [1,2] ✉

Deactivation of metal-based catalysts for vinyl chloride synthesis via acetylene hydrochlorination is often dictated by indispensable, catalytically-active carbon supports, but underlying mechanisms remain unclear. Carbon nitrides offer an attractive platform for studying them thanks to ordered structure and high N-content, which facilitates coking. Herein, we monitor the life and death of carbon nitride supports for Pt single atoms in acetylene hydrochlorination, demonstrating that specific N-functionalities and their restructuring cause distinct deactivation mechanisms. Varying polymerization and exfoliation degrees in pristine carbon nitrides (i.e., −NH$_x$ termination and N-vacancy concentrations), we establish graphitic and pyridinic N-atoms as C$_2$H$_2$ adsorption sites and pyridinic N-vacancies as coking sites through kinetic and spectroscopic analyses. Uniquely suited for probing point defects, operando electron paramagnetic spectroscopy, coupled to simulations, reveals that HCl drives depolymerization, by protonating heptazine-linking graphitic N-atoms, and generates graphitic N-vacancies, forming NH$_3$. These reduce C$_2$H$_2$ adsorption and promote radical polymerization into coke, respectively, without altering Pt atoms. Design guidelines to mitigate deactivation are discussed, highlighting the importance of tracking active functionalities in carbons.

Acetylene hydrochlorination is a key process for synthesizing vinyl chloride monomer (VCM, 13 Mton y$^{-1}$)[1,2]. From industrial toxic HgCl$_2$ catalysts to more sustainable metal alternatives (e.g., Pt, Au, Ru, Pd, and Cu) in distinct nanostructures, heterogeneous catalysts for this reaction share carbon supports as necessary components[3–8]. Activated and N-doped carbons have been widely employed, and also exhibit moderate intrinsic activity as metal-free catalysts[1,9,10]. In promising Pt single-atom catalysts (SACs), carbon supports were shown to act as C$_2$H$_2$ reservoirs, with chemisorption linked to structural characteristics such as functional group abundance and acidity. The corrosive reaction environment has hindered operando studies, so far leaving the role of Pt sites versus carbon in C$_2$H$_2$ activation unclear. Only recently, operando X-ray absorption spectroscopy (XAS) spectroscopy tracking metal atoms indicated that

[1]Institute for Chemical and Bioengineering, Department of Chemistry and Applied Biosciences, ETH Zürich, Vladimir-Prelog-Weg 1, 8093 Zürich, Switzerland. [2]NCCR Catalysis, Zürich, Switzerland. [3]Institute of Molecular Physical Science, Department of Chemistry and Applied Biosciences, ETH Zürich, Vladimir-Prelog-Weg 1, 8093 Zürich, Switzerland. [4]Institute of Chemical Research of Catalonia (ICIQ-CERCA), Avinguda Països Catalans 16, Tarragona, Spain. [5]Department of Physical and Inorganic Chemistry, Universitat Rovira i Virgili, Marcel·lí Domingo s/n, Tarragona, Spain. [6]Laboratory of Inorganic Chemistry, Department of Chemistry and Applied Biosciences, ETH Zürich, Vladimir-Prelog-Weg 1, 8093 Zürich, Switzerland. [7]Consiglio Nazionale delle Ricerche (CNR), Istituto Officina dei Materiali (IOM), Strada Statale 14 km 163.5, 34149 Basovizza, Italy. ✉e-mail: jpr@chem.ethz.ch

they exclusively bind HCl while complementary ex situ spectroscopy, $C_2H_2$ sorption, and computational investigations suggested that carbon supports activate acetylene[11]. Still, the amorphous nature of carbons hampers our understanding of their dynamic behavior during reaction. For example, while N-functionalities are known to induce deactivation through coking and subsequent blockage of metal sites[9], several potential deactivation mechanisms might happen simultaneously. The lack of characterization tools capable of (*i*) identifying the specific nature of the functionalities that drive acetylene polymerization into coke, (*ii*) monitoring carbon restructuring, and (*iii*) assessing its impact on catalyst reactivity severely limits our ability to discern and ultimately mitigate deactivation pathways.

Conventionally employed characterization techniques often face notable challenges when studying structurally-heterogeneous carbons containing light-scattering C-, N-, and O-atoms. For example, soft XAS faces penetration depth issues while its averaging nature complicates resolution of distinct species. While ex situ X-ray photoelectron spectroscopy (XPS) analyses suggested that acidic pyrrolic N-sites in N-doped carbons facilitate coke formation[3,12], conducting operando investigations under ambient pressure and in corrosive environments poses major practical challenges[13,14]. To date, information on carbon restructuring or deactivation mechanisms in acetylene hydrochlorination, and more generally reactions utilizing carbon-supported catalysts[15–18], remains largely experimentally inaccessible. Hence, mechanistic investigations mostly rely on density functional theory (DFT) simulations that often are not exhaustive in generating possible alternative structures[6,11]. In turn, the lack of experimental structural characterization of amorphous regions (e.g., defect types) hinders the identification and construction of accurate structural models[19,20]. Electron paramagnetic resonance (EPR) spectroscopy stands out as a powerful tool for investigating carbons under working conditions due to its ability to selectively detect paramagnetic species, including point defects, and probe light-scattering materials non-destructively[21,22]. Taking advantage of the intrinsic structural stability of Pt single atoms[3], a detailed understanding of dynamic processes could be obtained by investigating carbon-based supports with a more regular structure than amorphous carbons, such as carbon nitrides (CN). In acetylene hydrochlorination, CN showed suitability for supporting Au single atoms, which sinter during reaction[23], while their defect abundance can influence the interaction of reactants with the catalyst surface[24]. Furthermore, their high N-content leads to extensive coke formation[25]. These properties make CN a highly suitable platform to study distinct deactivation mechanisms and precisely identify the nature of the sites involved, providing key information to ultimately design stable catalysts.

Herein, we track CN supports for Pt single atoms during acetylene hydrochlorination, evidencing with high precision that specific N-functionalities and their restructuring lead to distinct deactivation mechanisms. Varying CN polymerization and exfoliation degrees (i.e., $-NH_x$ termination and N-vacancy concentrations)[26,27], we identify graphitic and pyridinic N-atoms as $C_2H_2$ adsorption sites and pyridinic N-vacancies as coking sites via kinetic and spectroscopic analyses. Operando EPR investigations reveal that HCl induces surface depolymerization and formation of graphitic N-vacancies. While Pt atoms remain unaltered, these restructuring phenomena reduce $C_2H_2$ adsorption and enhance coking, lowering activity, as revealed by operando EPR and supported by DFT mechanistic models. Finally, we discuss guidelines for designing carbon supports to resist deactivation. These results exemplify the need to understand the non-innocent role of carbon supports in catalyst deactivation.

## Results

### Platform of CN supports for Pt single atoms

CN supports are synthesized adapting previously reported protocols to vary polymerization and exfoliation degrees, regulating $-NH_x$ terminations and surface area, respectively (Fig. 1, Supplementary Table 1)[26]. Distinct scaffolds are derived from thermal treatment of melamine at different temperatures, i.e., 723, 773, and 823 K, controlling the polymerization degree and yielding linear melem oligomers (LMO), partially-polymerized CN (ppCN), and graphitic CN. These are subsequently thermally exfoliated, indicated by 'E' prefix in the sample code. Thereafter, Pt SACs are synthesized by incipient wetness impregnation with an aqueous solution of $H_2PtCl_6$ and thermal activation (473 K), and denoted as $Pt_{SA}$/support. Atomic metal dispersion is corroborated by the absence of Pt reflections in X-ray diffraction (XRD, Supplementary Fig. 1) and the visualization of isolated atoms by high-angle annular dark-field scanning transmission electron microscopy (HAADF-STEM, Fig. 1 and Supplementary Fig. 2). To quantitatively assess the metal nanostructure distribution, the HAADF-STEM images collected for each catalyst are analyzed by an atom-detection pipeline, which determines the distribution of nearest-neighbor distances (NND) based on advanced supervised and unsupervised methods (Supplementary Fig. 2)[28]. This approach leverages convolutional neural networks for pixel-wise metal center identification, combined with Gaussian mixture models to resolve overlapping features and identify low-nuclearity clusters. Although sporadic clusters are detected, the mean NND values for all catalysts exceed 0.32 nm, indicating distances larger than the dimer threshold, 0.24 nm, with $Pt_{SA}$/ECN showing a value around 0.5 nm. The cationic nature of platinum, primarily $Pt^{2+}$ and minor $Pt^{4+}$, is assessed by XPS, confirming its stabilization as single atoms by chloride ligands and interaction with N functionalities in the CN supports (Supplementary Fig. 3, Supplementary Tables 2–4). In fact, the formation of Pt nanoparticles, denoted as $Pt_{NP}$/ECN, requires a reductive treatment by hydrogen, which removes chloride ligands and prompts metal sintering, as visualized in HAADF-STEM analysis (Supplementary Fig. 4).

Detailed investigations on the CN structures are conducted. XRD analysis confirms their semi-crystalline nature (Supplementary Fig. 1)[26,29]. $Pt_{SA}$/CN exhibits an intense reflection at 27°, from graphite-like interlayer stacking of polymerized heptazine units, preserved in the exfoliated counterpart $Pt_{SA}$/ECN. In $Pt_{SA}$/LMO, diffraction peaks from 10 to 15° and from 25 to 30° (2θ) span melem-derived oligomers. In the exfoliated analog $Pt_{SA}$/ELMO, these peaks broaden, indicating the copresence of melem- and melon-based oligomers[29], reflecting partial polymerization during exfoliation. Complementary spectroscopic analyses provide deeper insights into the CN structures (Fig. 2). $^{13}C$ solid-state cross-polarization/magic angle spinning nuclear magnetic resonance (NMR) (CP/MAS NMR, Fig. 2a) spectroscopy corroborates the presence of two prevalent *C* species in $Pt_{SA}$/LMO and $Pt_{SA}$/ELMO, specifically one in aromatic non-protonated rings (156 ppm) and one linked to unreacted end $-NH_2$ groups (165 ppm)[30]. The same C species are observed in the highly polymerized counterparts $Pt_{SA}$/CN and $Pt_{SA}$/ECN, though a third contribution (163 ppm) appears from $-CNH$ species bridging the heptazine units[31]. Nevertheless, while stark differences in the peak width and position are noted between $Pt_{SA}$/LMO and $Pt_{SA}$/CN, spectral features become more similar in their exfoliated counterparts $Pt_{SA}$/ELMO and $Pt_{SA}$/ECN. Likewise, Fourier transform infrared (FT-IR, Fig. 2b) spectroscopy shows structural differences between $Pt_{SA}$/LMO and $Pt_{SA}$/CN in the stretching at $1100-1650$ cm$^{-1}$, corresponding to aromatic heterocycles, and the broad band at $3000-3300$ cm$^{-1}$, reflecting $-NH_x$ terminations (more prominent in $Pt_{SA}$/LMO)[32]; while spectral features become more similar between $Pt_{SA}$/ELMO and $Pt_{SA}$/ECN. Still, the support structure of these two catalysts is distinct owing to the different degrees of polymerization of the matrices, as corroborated by Raman spectroscopy (Supplementary Fig. 5). This is confirmed by ultraviolet-visible diffuse reflectance (UV-vis DRS) spectroscopy (UV-vis DRS, Fig. 2c) analysis, showing distinct melem- and melon-like polymerized structures for $Pt_{SA}$/ELMO and $Pt_{SA}$/ECN, respectively[29]. To further investigate the structure of the supports in these two Pt SACs, we have conducted soft

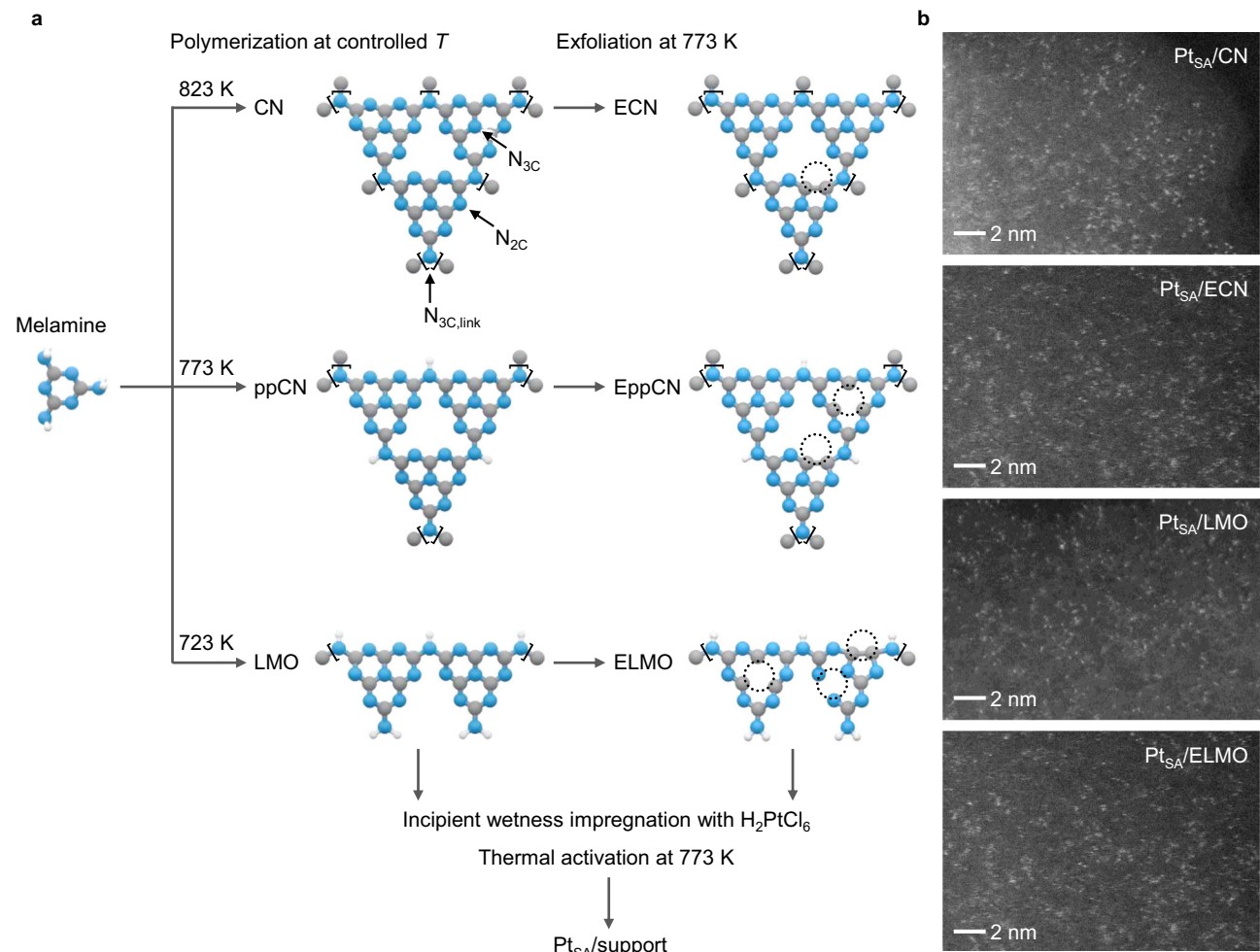

**Fig. 1 | Synthesis of carbon nitride supports for Pt single atoms. a** Scheme of synthesis approach and resulting structures of distinct carbon nitride supports for Pt SACs. These include graphitic carbon nitride (CN), partially-polymerized carbon nitride (ppCN), linear melem oligomers (LMO), and their exfoliated counterparts, indicated with 'E'. Three *N*-atom types are present: pyridinic N-atoms ($N_{2C}$), graphitic N-atoms ($N_{3C}$), and graphitic N-atoms linking three heptazine units ($N_{3C,link}$). Monomeric units are indicated with parentheses, while vacancies forming upon exfoliation are marked with dotted circles. Color code: blue N, gray C, white H. **b** HAADF-STEM images of selected carbon nitride-supported Pt SACs.

XAS analysis at the nitrogen K edge (Fig. 2d). Both the ECN and ELMO supports consist of polymeric units derived from heptazine rings, with varying degree of −NH$_x$ terminations and N-vacancies (*vide supra*, Fig. 1). In line with this and the bulk-averaging nature of XAS, the spectra of Pt$_{SA}$/ECN and Pt$_{SA}$/ELMO exhibit similar features. Specifically, in agreement with literature reports[33], we note three spectral contributions to the X-ray absorption near edge structure (XANES): at 399.6 eV (N1), 401.5 eV (N2), and 402.3 eV (N3). N1 is assigned to the N $1s \rightarrow \pi^*$ transition in aromatic $N_{2C}$-atoms of heterocyclic rings, $\pi^*$ (C = N −C); N2 to graphitic $N_{3C}$-atoms, $\pi^*$(N−3C); and N3 to sp$^3$ (potentially protonated) $N_{3C,link}$-atoms, $\pi^*$(N−C), respectively[33]. Nevertheless, the shape of the N1 contribution appears to be broader, shifting to higher energies, in Pt$_{SA}$/ELMO than Pt$_{SA}$/ECN. By comparison with reference materials featuring protonated N-atoms, dicyandiamide and mela-mine, which exhibit spectral contributions at 399.7 and 399.9 eV, respectively, the larger shoulder of the N1 contribution in Pt$_{SA}$/ELMO is tentatively attributed to higher protonation of the support, i.e., more −NH$_x$ terminations.

Still, exfoliation treatments enhance the formation of defects, which commonly used techniques cannot detect. Uniquely suited to probe point defects, EPR spectroscopy is employed. The continuous wave EPR (CW-EPR) spectrum of Pt$_{SA}$/ECN (Fig. 2e) at room tempera-ture (ca. 298 K) shows a narrow signal around $g = 2$, attributed to

paramagnetic point defects. By conducting a low-temperature (10 K) measurement at higher microwave attenuation power to better high-light spectral features, three main components in the CW-EPR spec-trum are identified through simulations (Fig. 2e, Supplementary Table 5): (*i*) a broader, moderately anisotropic signal, (*ii*) a more iso-tropic signal with a slightly lower *g*, and (*iii*) a minor narrow signal[21]. The latter is neglected because of its two orders of magnitude lower intensity. Possible paramagnetic atomic vacancies are considered to identify the first two components. The removal of a N-atom leads to a paramagnetic vacancy, unlike a C-atom. Since CN exhibit (*i*) pyr-idinic N-atoms in the triazine cavity ($N_{2C}$), (*ii*) graphitic N-atoms in the heptazine unit ($N_{3C}$), (*iii*) graphitic N-atoms linking three heptazine units ($N_{3C,link}$); their respective N-vacancies are investigated (Fig. 1a). As $N_{3C,link}$-vacancies would cause bond breaking and depolymerization (Fig. 1a), $N_{3C}$- and $N_{2C}$-vacancies are examined[34].

Preliminarily, the more anisotropic signal with a higher *g* is ascribed to more asymmetric $N_{2C}$-vacancies, while the virtually iso-tropic signal with a lower *g* is attributed to the more symmetric $N_{3C}$-vacancies, as supported by spin density plots using a plane-wave basis approach (Supplementary Fig. 6, Supplementary Table 6). These are further resolved in Q-band echo-detected field sweep spectra (Sup-plementary Fig. 7), showing an isotropic peak at 12,394 G and a lower-field anisotropic one at 12,385 G. Q-band 2-pulse electron spin echo

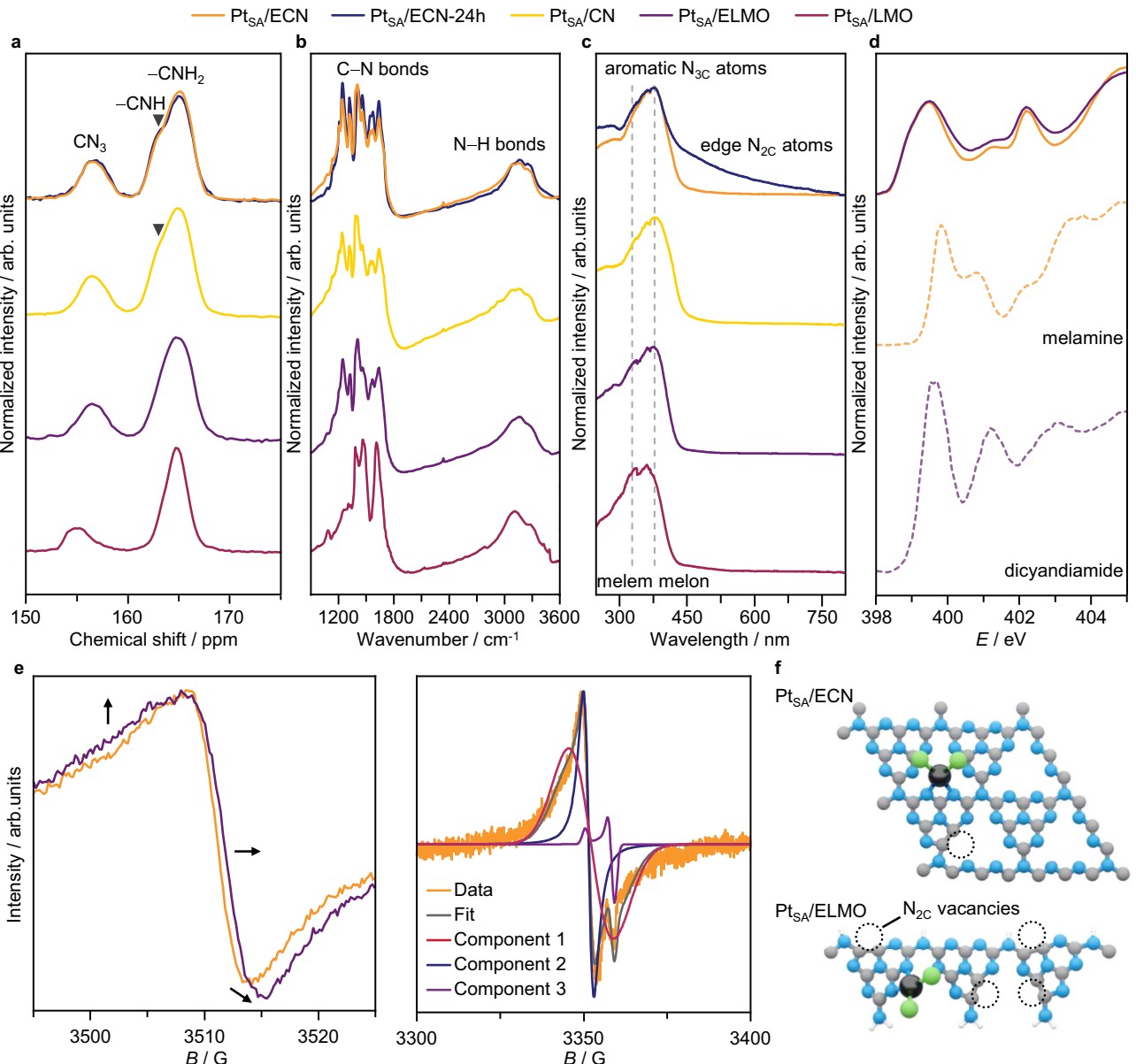

**Fig. 2 | Structure of the carbon nitride supports for Pt single atoms. a** $^{13}$C CP/ MAS NMR, **b** FT-IR, **c** UV-vis DRS, **d** N $K$-edge XANES, **e** CW-EPR, with experimental data (298 K and 20 dB, left; 10 K and 40 dB, right) and fitting components, spectra of selected carbon nitride-supported Pt SACs, as-prepared and after use in acetylene hydrochlorination, together with reference materials. Arrows in **e** mark an increasing number of $N_{2C}$-vacancies (**f**) Schematic representation of the difference in point defect abundance in $Pt_{SA}$/ECN and $Pt_{SA}$/ELMO, as observed by CW-EPR analysis. Color code: black Pt, blue N, green Cl, gray C, white H. Source data are provided as a Source Data file.

envelope modulation (ESEEM) experiments detect weak hyperfine couplings with nearby magnetic nuclei, revealing the structural environment of the paramagnetic defects. The ESEEM spectra at 12,394 G (showing both isotropic and anisotropic signals) and 12,385 G (showing only anisotropic signal) match (Supplementary Fig. 8), indicating that most peaks originate from $N_{2C}$-vacancies. Furthermore, DFT-computed hyperfine couplings for two tri-heptazine units, with one $N_{3C}$- and one $N_{2C}$-vacancy each (in a 1:1 ratio), show a more anisotropic and delocalized spin density distribution for $N_{2C}$-vacancies than $N_{3C}$ ones. The ESEEM spectrum simulated with these parameters closely matches the experimental $Pt_{SA}$/ECN spectrum (Supplementary Fig. 9).

The effect of thermal exfoliation on defect formation is further explored on a less polymerized support in $Pt_{SA}$/ELMO (Fig. 1). Its CW-EPR spectrum is similar to $Pt_{SA}$/ECN but more anisotropic, indicating a higher number of $N_{2C}$-vacancies, and shifted to higher fields corresponding to a lower isotropic $g$ value (Fig. 2e,f). However, $N_{2C}$-vacancies have a higher isotropic $g$ value than $N_{3C}$ ones, which suggests that another factor is at play. The decreasing $g$ value can be due to an increasing bandgap, leading to lower admixture of conduction and valence band states and consequently lower deviations from $g_e$ (free electron value). A lower bandgap can be related to lower polymerization degree. DFT simulations of small CN segments with varying numbers of condensed heptazine units, each with one $N_{3C}$-vacancy, show that the average $g$ value decreases as the unit number decreases (Supplementary Fig. 10). Thus, the key structural differences between ECN and ELMO are the higher concentration of $N_{2C}$-vacancies and lower polymerization in the latter.

## Reactivity of CN-supported Pt single atoms

The impact of the distinct structures of the CN supports on the catalytic activity of Pt single atoms for acetylene hydrochlorination is investigated (Fig. 3) by evaluating the VCM yield after 1 h on stream at typical operating conditions[3]. Despite similar speciation of the metal sites, the Pt SACs exhibit distinct initial activity, correlating with the $C_2H_2$ chemisorption capacity and surface area of the bare supports. Pt SACs on non-exfoliated, low surface area supports are virtually inactive, while exfoliated supports yield active catalysts. Unlike moderately active Pt$_{SA}$/ELMO, Pt$_{SA}$/ECN shows high activity, attributed to its higher surface area and polymerization degree. This results in fewer $-NH_x$ terminations and $N_{2C}$- vacancies (*vide supra*, Fig. 2e), indicating these sites are not involved in catalyzing VCM formation. To decouple the effect of the surface area from that of the polymerization degree, we synthesize and test a series of Pt SACs supported on ELMO, EppCN,

and ECN with constant metal-content-to-surface-area ratio (i.e., metal density, 75 $\mu mol_{Pt}$ m$^{-2}$). The catalyst mass, and thus space velocity, is varied to maintain a constant reactant flow rate per metal site (Supplementary Fig. 11). In line with the fewer $-NH_x$ terminations and $N_{2C}$-vacancies, the ECN-supported Pt SAC shows higher activity, followed by the EppCN- and ELMO-supported analogs. These findings denote that the CN properties govern $C_2H_2$ binding and thus catalytic activity, highlighting the key role carbon-based supports play in fulfilling the catalytic cycle. Still, $C_2H_2$ and HCl reaction orders derived for Pt$_{SA}$/ECN, 0.41 and 0.24 (Supplementary Fig. 12), respectively, are lower than 1, marking that both reactants participate in the catalytic cycle in their adsorbed state[5]. Consistently, bare CN without Pt single atoms are virtually inactive (Supplementary Table 1), as HCl cannot be activated to form VCM. Beyond catalytic activity, the properties of carbon-based supports also influence catalyst stability. For instance, high *N*-content is known to promote the formation of coke deposits[9,25]. In line with this, prominent deactivation is observed over 24 h on stream (Fig. 4a). Thermogravimetric analysis (TGA) confirms extensive formation of coke deposits, deriving from polymerization of $C_2H_2$ (Supplementary Fig. 13), linking the amount of coke formation across Pt$_{SA}$/ECN, Pt$_{SA}$/EppCN, and Pt$_{SA}$/ELMO with their respective $C_2H_2$ adsorption capacity of their bare supports. Additionally, deactivation rates can be correlated with the isotropic *g* value of the paramagnetic centers in the CN supports with varying polymerization degrees, in both the as-prepared Pt SACs and after use in acetylene hydrochlorination for 24 h (Fig. 4a, Supplementary Fig. 14). The differences between the *g* values reflect the differences in activity at the beginning of the reaction and after 24 on stream. This is further confirmed by catalytic tests, where varying the gas-hourly space velocity results in comparable VCM yield across catalysts (Supplementary Fig. 15), also showing that deactivation progresses until near inactivity after *ca.* 48 h on stream. This correlation highlights that the more $-NH_x$ terminations and $N_{2C}$-vacancies are present (*vide supra*), the more prominent the deactivation is. This can be attributed to support restructuring or coking having a greater impact on CN matrices with fewer active N-sites. Deeper insights into the dynamic behavior of Pt$_{SA}$/ECN are gained by analyzing the reactor outlet stream by time-resolved mass spectroscopy (Fig. 4b). The signal of both $C_2H_2$ and HCl gradually increase over time on stream, while the VCM signal increases, denoting catalyst deactivation. Nevertheless, the $C_2H_2$ signal initially plateaus (Supplementary Fig. 16), suggesting consumption toward VCM formation, but also coke formation.

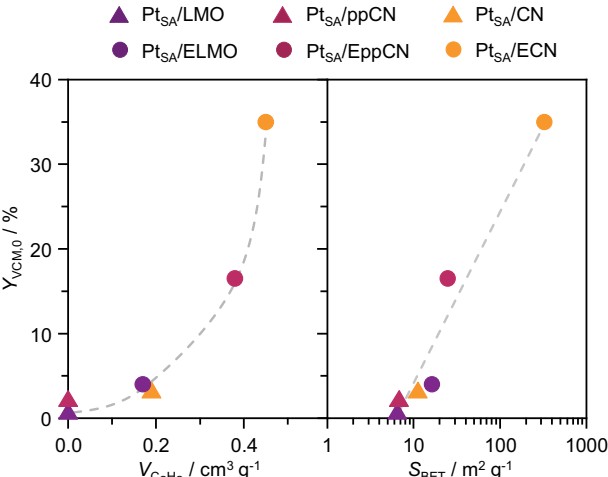

**Fig. 3 | Activity descriptors for carbon nitride-supported Pt single atoms.** Initial activity, expressed as VCM yield after 1 h on stream, $Y_{VCM,0}$, of carbon nitride-supported Pt SACs as a function of $C_2H_2$ chemisorption capacity, $V_{C_2H_2}$ (left), and surface area, $S_{BET}$ (right), of the bare carbon nitrides. Source data are provided as a Source Data file.

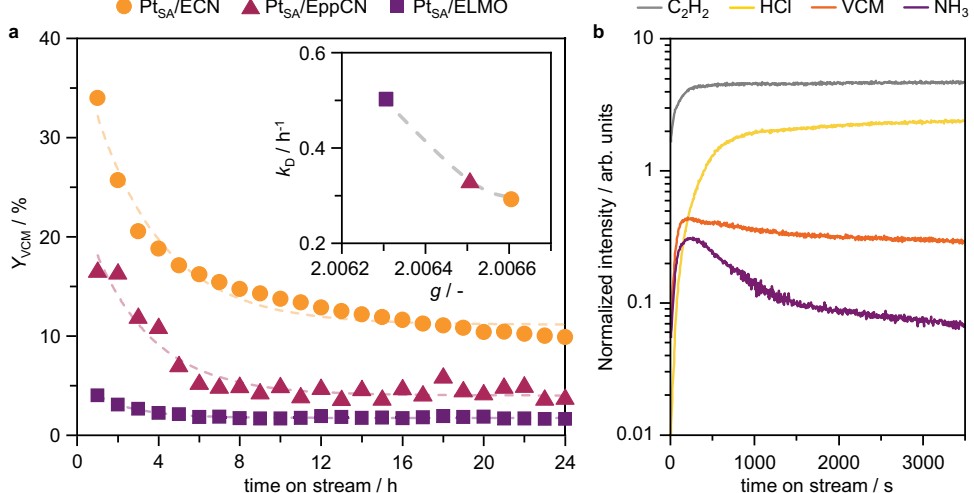

**Fig. 4 | Deactivation of carbon nitride-supported Pt single atoms. a** Catalytic performance, $Y_{VCM}$, of selected carbon nitride-supported Pt SACs over time on stream with respective deactivation constants, $k_D$, as determined by exponential regression, and average *g* factor, as determined by CW-EPR, shown in the inset.

**b** Time-resolved product analysis over Pt$_{SA}$/ECN by mass spectroscopy, monitoring m/z 26 ($C_2H_2$), 36 (HCl), 62 (VCM), and 17 (NH$_3$). Source data are provided as a Source Data file.

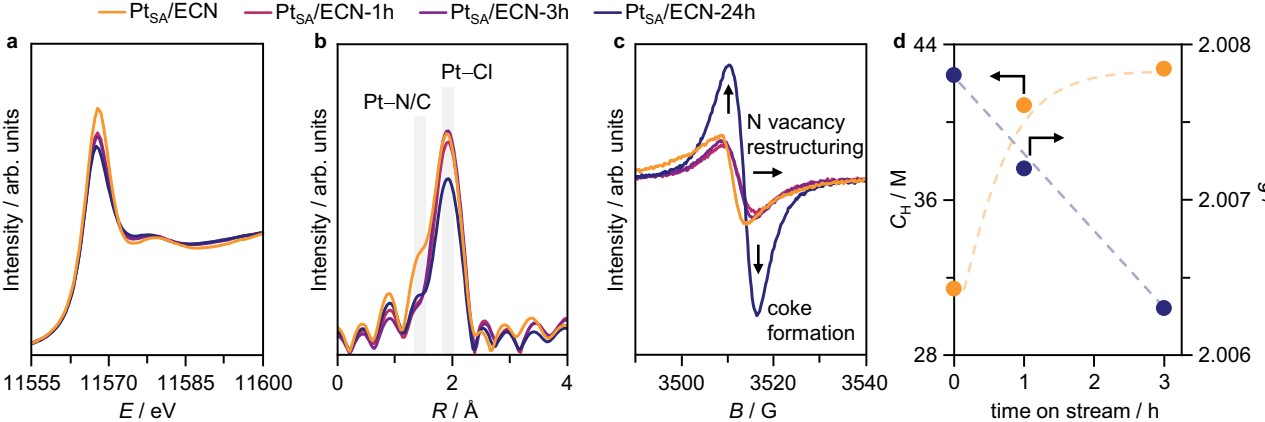

**Fig. 5 | Ex situ characterization of carbon nitride-supported Pt single atoms after use in acetylene hydrochlorination.** Pt $L_3$ edge (**a**) XANES, **b** EXAFS, and **c** CW-EPR spectra of $Pt_{SA}$/ECN, as-prepared and after use in acetylene hydrochlorination. **d** Average $g$ factor and ih-RIDME-determined local proton concentration around the N-vacancies, $C_H$, as a function of time on stream, as measured in CW-EPR analysis. Source data are provided as a Source Data file.

Interestingly, another product, $NH_3$, is detected. Its formation gradually decreases over time, with kinetics that seem to align with HCl consumption, suggesting that $NH_3$ may be produced via protonation of the N-functionalities in the CN support (*vide infra*). The central question remains whether deactivation mechanisms, such as active site restructuring and/or coking, impact the Pt atoms or the carbon functionalities.

Deactivation *via* metal sintering during reaction conditions is ruled out by HAADF-STEM analysis of $Pt_{SA}$/ECN after reaction (Supplementary Fig. 17), showing atomic metal dispersion. Nevertheless, sporadic clusters might be present from the as-prepared catalysts. To probe the effect of metal nanostructure on catalyst performance, Pt nanoparticles supported on ECN, $Pt_{NP}$/ECN, are tested (*vide supra*, Supplementary Figs. 4 and 18). Owing to their reduced ability to activate HCl[3], Pt nanoparticles exhibit a three-fold lower activity than the $Pt_{SA}$/ECN counterpart. Still, catalyst deactivation remains prominent, suggesting that the CN support regulates this process rather than distinct Pt species. Detailed insights into the oxidation state and coordination environment of the Pt atoms before and throughout the reaction can be gained by XAS, through analysis of the XANES and extended X-ray absorption fine structure (EXAFS, Fig. 5a,b). Metal sites in the as-prepared $Pt_{SA}$/ECN exhibit a high oxidation state, resembling $Pt^{4+}$, and a prominent Pt−Cl contribution (coordination number, CN = 3.2, Supplementary Fig. 19, Supplementary Table 7) that derive from the $H_2PtCl_6$ precursor (Supplementary Fig. 20). Notably, these are further slightly chlorinated after 1 and 3 h on stream (CN = 3.3 and 3.4, respectively, Fig. 5b), when catalyst deactivation is most prominent (Fig. 4). This is consistent with the HCl-activating function of the metal sites and is counteracted by a minor loss in the Pt−N/C contribution (CN = 0.8 and 0.7 after 1 and 3 h on stream, respectively), linked to slight reduction in the whiteline intensity in the XANES spectra (Fig. 5a). No interactions between $C_2H_2$ and the Pt single atoms are noted[11]. These results point to the deactivation by coking affecting the CN support rather than the Pt single atoms. A slight decrease and increase in the Pt−Cl and Pt−N/C contributions, respectively, are noted only after 24 h on stream (Pt−Cl CN = 2.6 and Pt−N/C CN = 1.1, Fig. 5b), when the deactivation has fully taken place, suggesting that Pt atoms might eventually suffer from minor blockage by coke deposits formed over the support. To clarify the influence of each reactant on the Pt single atoms, $Pt_{SA}$/ECN is exposed separately to HCl and $C_2H_2$ at 473 K and analyzed by XAS (Supplementary Figs. 19 and 20, Supplementary Table 7). When exposed to HCl, the Pt single atoms extensively chlorinate (Pt−Cl CN = 5.2) as no chloride ligands are consumed to form VCM. In contrast, exposure to $C_2H_2$ results in nearly full

dechlorination, with prominent Pt−$C_2H_2$ interactions appearing in the EXAFS spectrum (Pt−N/C = 3.3 for long-bonding contributions at 2.04 Å)[11]. These findings confirm the role of Pt single atoms in activating HCl during reaction, rather than primarily binding $C_2H_2$.

Characterization of $Pt_{SA}$/ECN after use in acetylene hydrochlorination by $^{13}C$ CP/MAS NMR spectroscopy indicates no stark structural changes of the CN matrix (Fig. 2a,b). In FT-IR spectra, slight changes are noted between 1600−1800 $cm^{-1}$, suggesting minor changes in C−N bonds, while the broad band at 3000−3300 $cm^{-1}$ remains virtually unaltered post-reaction, identifying −$NH_x$ terminations as spectator species. Analysis by UV-vis DRS shows the appearance of a contribution at ca. 470 nm, reflecting the buckling of $N_{2C}$-atoms that results from their interaction with the reaction mixture (Fig. 2c). Consistently, XPS analysis reveals catalyst surface chlorination, including the ECN support (Supplementary Fig. 3, Supplementary Table 2). To further investigate carbon-$C_2H_2$ interactions and complement $C_2H_2$ chemisorption measurements (*vide supra*), temperature-programmed desorption analyses of acetylene coupled to mass spectrometry ($C_2H_2$-TPD-MS) are conducted over the ECN support (Supplementary Fig. 21). Desorption of $C_2H_2$ was observed between 350 and 500 K. However, from 500 K onward, the CN support begins to undergo thermal decomposition, which interferes with the detection of desorbed $C_2H_2$. Specifically, the $C_2H_2$ MS signal at 26 $m/z$ overlaps with that of cyanide, a byproduct of the decomposition of ECN, as confirmed by the peak detected when analyzing the CN support when flowing He only, in the absence of $C_2H_2$. These findings highlight the susceptibility of CN to structural changes.

To investigate restructuring phenomena in the ECN support during the reaction, ex situ CW-EPR analysis is performed on $Pt_{SA}$/ECN after 1 and 3 h on stream (Fig. 5c). Both spectra show a progressive shift to higher fields compared to the as-prepared catalyst. This shift can be attributed to increased $N_{3C}$:$N_{2C}$-vacancy ratio (*vide supra*), as suggested by the more isotropic lineshape, and/or depolymerization. However, the relatively large shift and modest lineshape changes indicate that both effects likely contribute. This is confirmed by the changes observed in the X-band and Q-band 2-pulse ESEEM spectra (Supplementary Fig. 22), showing quenching of $N_{2C}$-vacancies (*vide supra*) during reaction. To further explore depolymerization effects, we employ different pulsed EPR methods.

As depolymerization occurs, the heptazine network breaks by $N_{3C,link}$ atom protonation. Therefore, the concentration of protons that are weakly coupled with unpaired electrons in the reaction-induced $N_{3C}$-vacancies increases. Hyperfine couplings with protons are known to be the main contribution to the Hahn echo decay[35]. By comparing

the decay traces for the as-prepared $Pt_{SA}$/ECN and after 1 and 3 h on stream (Supplementary Fig. 23), we observe a significant increase in the decay rate. These decays are independent of microwave power attenuation, indicating that instantaneous diffusion effects and, consequently, electron-electron spin interactions are negligible. The low intensity and slow decay rate suggest the observed changes are entirely ascribable to increased proton concentration leading to depolymerization. This is confirmed by stimulated echo decay experiment, equivalent to intermolecular hyperfine relaxation-induced dipolar modulation enhancement (ih-RIDME, Supplementary Figs. 24 and 25, Supplementary Table 8)[36–39], a newly-developed method that analyzes echo decays and provides a quantitative estimate of local proton concentration. The latter increases rapidly over time (Fig. 5d) while the $g$ value decreases, linking depolymerization with catalyst deactivation (Fig. 4). Nevertheless, as no coke radicals are noted, these results suggest that a distinct deactivation mechanism is at play (*vide infra*). At last, the CW-EPR spectrum of $Pt_{SA}$/ECN after 24 h on stream exhibits a significantly stronger signal with a much faster spin echo decay (Fig. 5d), resembling that typically observed for radicals in coke[40]. This is confirmed by X-band 2-pulse ESEEM and hyperfine sublevel correlation (HYSCORE) spectrum (Supplementary Figs. 22 and 26), which closely matches the spectrum of radicals in coke on activated carbon[11]. Interestingly, a reaction-induced low-field shift of the CW-EPR signal is observed for the bare ECN support upon use in acetylene hydrochlorination for 24 h, also undergoing deactivation over time on stream despite the lack of Pt sites (Supplementary Figs. 27 and 28). This indicates that the CN support can interact with the reactants and restructure independently of the metal sites, which further underscores the key role of the carbon support in regulating the performance of SACs in acetylene hydrochlorination.

## Reaction-induced restructuring and deactivation of the CN support

To investigate the role of CN supports in catalytic deactivation, their structural changes are monitored, for the first time, in $Pt_{SA}$/ECN by operando CW-EPR (Fig. 6a–c). Provided the corrosiveness of HCl, dilute reactant concentrations are employed to ensure the equipment and personnel safety, while the reaction temperature is maintained at 473 K. Similarly to ex situ analyses, we observe a gradual high-field shift of the signal, indicating N-functionality restructuring. This phenomenon begins upon ramping the temperature up to 473 K, under Ar, and reaches equilibration (Fig. 6b), indicating that it is partially induced by thermal effects. Upon feeding the reactants, the restructuring process starts again and correlates with consumption of $C_2H_2$, since the signal in the mass spectrometer plateaus until the line shift in the CW-EPR signal is completed (Fig. 6b,c). As the restructuring process stops, the $C_2H_2$ signal increases steeply over time (Fig. 6c), which is unmatched by the HCl one, which agrees with the results of product analysis by mass spectrometry at the reactor outlet during testing at high reactant concentrations (*vide supra*, Supplementary Fig. 16). This indicates that deactivation is caused by restructuring of the CN support, reducing $C_2H_2$ consumption (i.e., activation).

In situ experiments are conducted to disentangle the effect of temperature, $C_2H_2$, and HCl (Fig. 6d). First, in situ CW-EPR analysis of $Pt_{SA}$/ECN spectra collected at 298 K in Ar after heating to reaction temperature corroborates that $N_{2C}$-to-$N_{3C}$ vacancy restructuring is partially temperature-induced (i.e., thermodynamically driven). Then, in situ CW-EPR experiments are conducted on $Pt_{SA}$/ECN exposed to the individual reactants at room temperature (ca. 298 K). $C_2H_2$ only induces a slight decrease in the signal intensity. This indicates quenching of $N_{2C}$-vacancies that is ascribable to $C_2H_2$ adsorption and/or coke formation (Supplementary Fig. 13). Since coke radicals are not observed, consistently with the short time-on-stream and dilute reactant concentrations, the former explanation appears more likely. Contrarily,

HCl causes a clear high-field shift, revealing HCl-induced formation of $N_{3C}$-vacancies and depolymerization, by $N_{3C,link}$ protonation.

Guided by dynamic structures of the CN support unveiled by operando CW-EPR, DFT simulations are conducted to propose mechanistic pathways for the restructuring of the CN matrix and the interaction of reactants with both metal sites and the support. The models were based on polymerized heptazine frameworks, explicitly representing the distinct nitrogen environments ($N_{2C}$, $N_{3C}$, and $N_{3C,link}$) and introducing controlled nitrogen vacancies to capture key defect structures observed experimentally. This approach preserves the essential local coordination features while offering a tractable representation of the extended CN network. Owing to the crystallinity of CN, $N_{2C}$-functionalities in the triazine cavities are unambiguously identified as the Pt atom anchoring sites[26,41]. XAS analysis points to the presence of dichlorinated $PtCl_2$ species, with reduced affinity for $C_2H_2$[11]. This is further supported by DFT analyses probing the competitive adsorption of HCl and $C_2H_2$ over pristine Pt atoms, featuring two chloride ligands and a two-fold coordination with the support, and working Pt sites, which are bichlorinated and coordinated one- or two-fold with the CN support presenting $N_{3C}$-vacancies (Supplementary Fig. 29). Simulations show $C_2H_2$ is hindered in the presence of HCl (by up to 0.26 eV) or even fully inhibited by chloride ligands. Bader charge analysis of $PtCl_2$ before and after HCl adsorption shows minimal electronic changes ($< 0.18|e^-|$, Supplementary Table 9), aligning with XPS characterization of the stable oxidation state in $Pt_{SA}$/ECN as-prepared and after use in acetylene hydrochlorination for 24 h (Supplementary Fig. 3), and underscoring the role of CN in stabilizing the Pt sites' electronic properties via metal-support electron distribution. Still, the CN matrix undergoes restructuring, as evidenced by CW-EPR. Starting from the pristine material (**a1** in Fig. 7a), $PtCl_2$ species can activate HCl, acquiring another chloride ligand while that may result in loss in coordination with support, as suggested by XAS analysis (*vide supra*). Hence, chlorinated atoms with both one- and two-fold coordination with the CN support are investigated. The H atom is transferred to a neighboring $N_{2C}$-atom in the triazine cavity (−0.72 and −0.25 eV, respectively, **a2** in Fig. 7a). The energy profile of subsequent restructuring steps is not significantly influenced by the Pt atom coordination, as it pertains only to the matrix, and only results for $PtCl_3$ species coordinated one-fold with the support are discussed in detail for simplicity. The resulting −NH functionality can mediate the activation of another HCl molecule by forming a −$NH_2$ species while the Cl atom binds a proximal *C* atom, bound to a $N_{3C,link}$-atom (−0.09 and 0.09 eV, **a3** in Fig. 7a). This breaks the planarity of the surface CN layer as the formed −$NH_2$ species forms H-bonds with the underneath layer, migrating toward it. The exergonic activation of a third HCl molecule (−1.22 and −1.02 eV, **a4** in Fig. 7a) generates a $NH_3$ molecule that can be released into the gas phase (0.81 and 0.25 eV, **a5** in Fig. 7a), in line with the detection of $NH_3$ by mass spectrometry (Fig. 4b), leaving a $N_{2C}$-vacancy and locally restructuring the heptazine unit (forming a five-membered ring). In turn, the Cl atom stabilizes over $N_{3C,link}$-neighboring C atom, thus inducing the breaking of the $NH_3$-C bond leading to dynamic heptazine buckling. The $N_{2C}$-vacancy migrates towards the center of the heptazine forming a $N_{3C}$-vacancy (0.50 and 0.27 eV, **a6** in Fig. 7a, Supplementary Fig. 30). This is accompanied by changes in the coordination of Cl species, which are known to be mobile, across the matrix[42–44]. $N_{3C}$-vacancy formation shows an unpaired electron confined within the heptazine unit, consistent with the radical nature of this defect observed by CW-EPR (Fig. 6).

Next, $C_2H_2$ adsorption over CN is investigated. While in a pristine defect-free structure $C_2H_2$ adsorption Gibbs free energies at the $N_{2C}$, $N_{3C}$, and $N_{3C,link}$ sites are thermoneutral (Fig. 7b, Supplementary Fig. 31, Supplementary Table 10), when corresponding N-vacancies are generated $C_2H_2$ adsorption becomes highly exergonic to −3.63, −1.04, and −5.81 eV, respectively. Upon adsorption, $C_2H_2$ forms a radical species, which can initiate polymerization into coke, consistent with CW-EPR

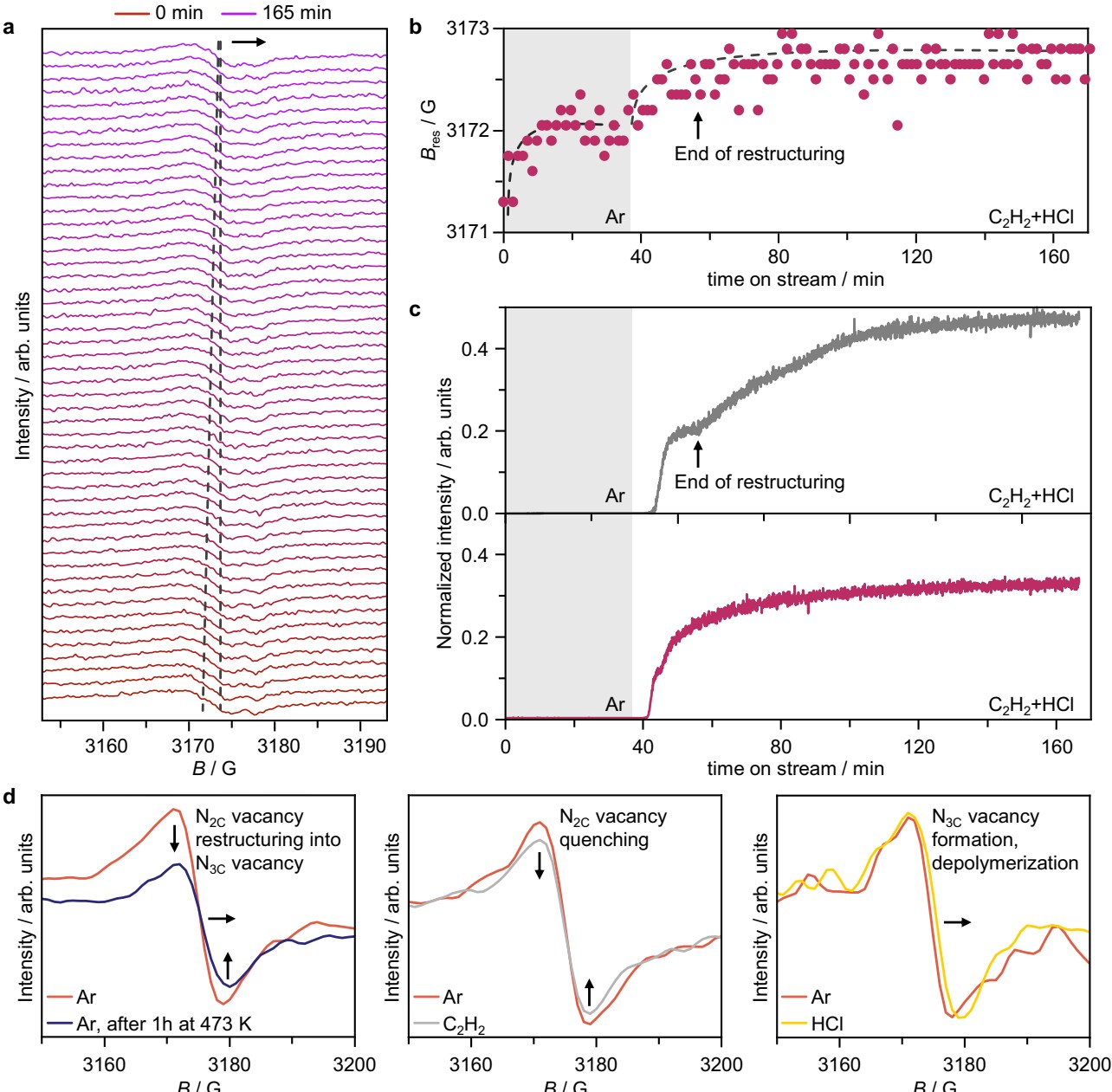

**Fig. 6 | Reaction-induced restructuring of the carbon nitride support. a** Operando CW-EPR spectra collected at 473 K of $Pt_{SA}$/ECN with corresponding **b** changes in the position of the maximum EPR absorption, $B_{res}$, over time, and **c** time-resolved $C_2H_2$ (gray) and HCl (purple) analysis by mass spectrometry. **d** in situ CW-EPR spectra collected at 298 K of $Pt_{SA}$/ECN, as-prepared and after heating to reaction temperature (i.e., 473 K, left), as well as during exposure to only $C_2H_2$ or HCl at 298 K (middle and right, respectively). Source data are provided as a Source Data file.

analyses (Figs. 5c and 6b,c). Additionally, in situ CW-EPR also evidences HCl-induced protonation of $N_{3C,link}$-atoms (Fig. 6d), inducing surface depolymerization. To investigate its impact on $C_2H_2$ activation, DFT analyses are conducted and reveal that depolymerization leads to reduced $C_2H_2$ adsorption (Fig. 7c, Supplementary Table 10). This surface disruption hampers the catalytic cycle and contributes to the deactivation of CN-supported Pt SACs.

## Discussion

This study provided an advanced understanding of working carbon-based supports and their role in catalyst deactivation—a central aspect for practical applicability often overlooked in academic research—for platinum-catalyzed acetylene hydrochlorination. Leveraging the

stability of Pt single atoms, this was achieved by (*i*) selecting CN as supports with ordered structure and high N-content that promotes coking, and (*ii*) employing EPR, utilizing its unique suitability for probing light-scattering materials and radical point defects (i.e., vacancies). A platform of CN supports with varying polymerization and exfoliation degrees was generated to correlate resulting structural properties and specific N-functionalities with catalytic activity and deactivation trends (Fig. 8). High surface area and low content of spectator $-NH_x$ terminations (i.e., high polymerization) were linked to high activity, while exfoliating low-polymerized supports led to pyridinic *N*-vacancy formation, promoting coking. Ex situ XAS and HAADF-STEM analyses revealed no structural changes in Pt atoms after reaction, while other spectroscopic techniques (NMR, FT-IR, UV-vis

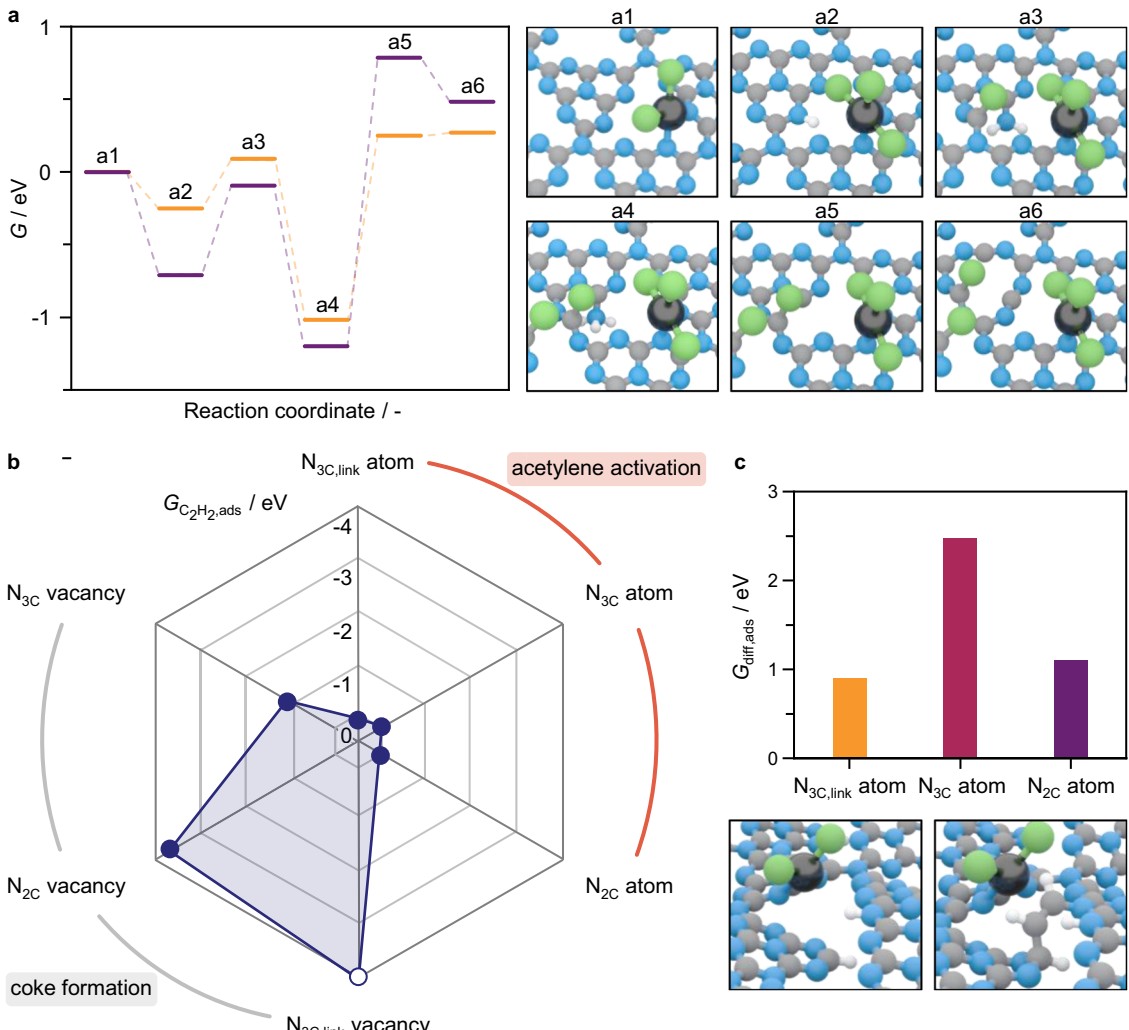

**Fig. 7 | Restructuring mechanism of the carbon nitride support and impact on reactivity. a** Gibbs free energy, $G$, profile of the HCl-induced formation of $N_{3C}$-vacancy in the ECN support for Pt SACs, and related structural representations. Two cases are considered: chlorinated Pt atoms one- (purple) and two-fold (orange) coordinated with the carbon nitride support, while structural representations are shown for the former. **b** Adsorption Gibbs free energy of $C_2H_2$, $G_{C_2H_2,ads}$, over distinct N-atoms and N-vacancies in the pristine ECN support for Pt SACs. The $G_{C_2H_2,ads}$ over $N_{3C,link}$-vacancy, marked by a hollow symbol, is below −4 eV, as the simulated model is a 2×2 cell that imposes non-physical constraints. This limits system relaxation, causing artificially too high vacancy structure energies and thus an overstabilized adsorbed $C_2H_2$ molecule. **c**, Difference in adsorption Gibbs free energy of $C_2H_2$, $G_{diff,ads}$, after HCl-induced depolymerization by $N_{3C,link}$ protonation. Structural representations in (**c**) show the depolymerized ECN before (left) and after (right) $C_2H_2$ adsorption on the $N_{2C}$-atom. Color code in structural representations in **a** and **c**: black Pt, blue N, green Cl, gray $C$, white H. Source data are provided as a Source Data file.

DRS, and XPS) indicated support restructuring. To gain deeper insights, working carbon-based supports were tracked for the first time by operando EPR. Far from being static, CN underwent HCl-induced generation of graphitic $N$-vacancies and surface depolymerization. Guided by magnetic parameters extracted from operando EPR spectra, DFT simulations proposed restructuring mechanisms consistent with experimental observations (Supplementary movie 1). The formation of EPR-active $N_{2C}$-vacancy point defects proceeds via ammonia elimination, promoting local reorganization within a heptazine unit. These defects are mobile and thermodynamically driven to form EPR-active $N_{3C}$-vacancies. Furthermore, their radical nature makes $N$-vacancies responsible for $C_2H_2$ polymerization and coke formation. Alternatively, HCl can react with and break the C−N bonds linking heptazine units, generating −$NH_x$ terminations and inducing layer buckling. The latter mechanism is long-range in nature and is responsible for the depolymerization of the carbon mesh. From this detailed mechanistic analysis, strategies to devise more resistant carbons can emerge. For instance, the basicity of N-functionalities can regulate HCl uptake, making precise control of acid-base properties essential. Additionally, reducing the polarity of C−N or C−$ZZ$ ($ZZ$ = O, S, P, or other functionalities) bonds can limit activation of polar molecules such as HCl, therefore increasing long-term materials stability. Finally, structures like CN, where monomers are linked by a single, labile C−N bond, can suffer from long-range structural disruption. To address this, strategies should focus on reinforcing chemical bonds in the matrix, for example through cross-linking. Beyond acetylene hydrochlorination, this work underscores the importance of elucidating deactivation mechanisms and how carbon supports can trigger them, to devise effective mitigation strategies.

## Methods
### Catalyst preparation
LMO, ppCN, and polymeric CN were prepared by calcination of melamine (8 g) at 723, 773, and 823 K. Exfoliated LMO, ppCN, and CN

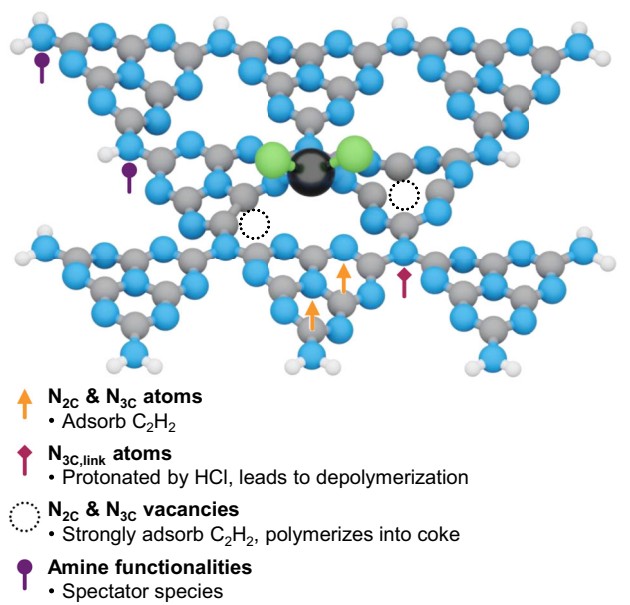

**N₂C & N₃C atoms**
- Adsorb C₂H₂

**N₃C,link atoms**
- Protonated by HCl, leads to depolymerization

**N₂C & N₃C vacancies**
- Strongly adsorb C₂H₂, polymerizes into coke

**Amine functionalities**
- Spectator species

**Fig. 8 | Catalytic role of N-functionalities in carbon nitride supports.** Summary of the catalytic function of distinct N-functionalities in carbon nitride supports for Pt SACs. Color code: black Pt, blue N, green Cl, gray C, white H.

(ELMO, EppCN, ECN) were obtained via thermal exfoliation of the powdered bulk materials (<0.3 mm) at 773 K in static air. The Pt SACs (SACs, nominal metal loading 1 wt%) were prepared via an incipient wetness impregnation method of the CN support with an aqueous solution of $H_2PtCl_6$, followed by thermal activation at 473 K in static air. Additionally, a nanoparticle-based catalyst supported on ECN was obtained via thermal activation in a reducing atmosphere (50/50 vol% $H_2$/He). Further details on the catalyst synthesis and the preparation of the carbon supports are provided in the Supplementary Methods.

### Catalyst characterization

Multiple techniques were employed to characterize the catalytic materials. The porous properties of the carbon supports were assessed by $N_2$ sorption at 77 K. The metal dispersion was assessed through XRD and high-angle annular dark-field scanning transmission electron microscopy (HAADF-STEM) combined with an atom detection inference pipeline to assess NND between metal centers and their mean value (<NND). The composition and chemical state of the metal atoms and the carbon supports were evaluated by XPS. The metal oxidation state and coordination environment of as-prepared Pt SACs and after use in acetylene hydrochlorination were evaluated by XAS, by XANES and extended X-ray absorption fine structure (EXAFS), respectively. Structural assessment of the CN supports in as-prepared Pt SACs and after use in acetylene hydrochlorination was assessed by NMR, Fourier-transform infrared (FT-IR), (UV-vis DRS), and soft XAS spectroscopies. CN restructuring during reaction was monitored by operando EPR spectroscopy. Coke deposits on the catalysts after use in acetylene hydrochlorination were quantified by TGA. All characterization techniques and procedures are detailed in the Supplementary Methods.

### Catalytic evaluation

The hydrochlorination of acetylene was evaluated at atmospheric pressure in a continuous-flow fixed-bed reactor set-up, as described elsewhere[3]. In a typical test, the catalyst ($m_{cat}$ = 0.25 g) was loaded in the quartz reactor and heated in a He flow to the desired bed temperature ($T_{bed}$ = 473 K). After stabilization for at least 15 min, the reaction mixture (40 vol% $C_2H_2$, 44 vol% HCl, and 16 vol% Ar) was fed at a total volumetric flow of $F_T$ = 7.5 cm³ min⁻¹, employing a high gas

hourly space velocity based on acetylene, $GHSV(C_2H_2)$ = 325 h⁻¹. Reactants and products, including the yield of vinyl chloride, the carbon mass balance, and mass and heat transfer limitations were evaluated according to the protocols described in the Supplementary Methods.

### Computational methods

DFT simulations were performed using the Vienna Ab initio Simulation Package, as detailed in the Supplementary Information. Projector augmented wave core potentials a cutoff energy of 450 eV and the PBE-D3 functional[45-49]. Transition states were located by using climbing image nudge elastic band and verified via frequency calculations[50]. The ECN support was modeled as a heptazine (2 × 2) supercell of four layers with the bottom one fixed to the bulk configuration, considering six coordination sites: (*i*) pyridinic N-atoms in the triazine cavity ($N_{2C}$), (*ii*) graphitic N-atoms in the heptazine unit ($N_{3C}$), (*iii*) graphitic N-atoms linking three heptazine units ($N_{3C,link}$), and (*iv–vi*) their respective vacancies. $Pt_{SA}$/ECN was modeled by placing $PtCl_2$ moieties on $N_{2C}$ sites. All computed structures can be retrieved from the ioChem-BD database[51]. The structural parameters (*g* and *A* tensors) for the analysis of experimental CW-EPR and 2-pulse ESEEM spectra were determined with Kohn-Sham DFT, using a B3LYP functional with a spin-unrestricted shell, a DGTZVP basis set for triazine radicals and a 6–31 G for extended CN systems in the Gaussian and Orca software[52,53].

### Data availability

The experimental and computational data generated in this study have been deposited in the Zenodo database under accession code 15304855 and in the ioChem-BD database (https://doi.org/10.19061/iochem-bd-1-352). The experimental and computational data generated in this study are provided in the Supplementary Information and Source Data file. Source data are provided with this paper.

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

## Acknowledgements

This study was created as part of NCCR Catalysis (grant number 225147), a National Center of Competence in Research funded by the Swiss National Science Foundation. The Spanish Ministry of Science and Innovation is acknowledged for financial support (PID2021-122516OB-I00 and Severo Ochoa Grant MCIN/AEI/10.13039/501100011033CEX2019-000925-S) and the Barcelona Supercomputing Center-MareNostrum (BSC-RES) for providing generous computer resources. A.R.-F. and J.M.G.-A acknowledge funding from the Generalitat de Catalunya and the European Union under Grants 2023 FI-3 00027 and 2024 FI-1 00437, respectively. F.B. acknowledges funding from the European Union - NextGenerationEU, Mission 4, Component 2, under the Italian Ministry of University and Research

(MUR) National Innovation Ecosystem grant ECS00000041 - VITALITY - CUP B43C22000470005. The Swiss-Norwegian (SNBL, ESRF) and BACH (Elettra) beamlines are acknowledged for provision of beamtime and its staff for invaluable support. The Scientific Center for Optical and Electron Microscopy (ScopeM) at the ETH Zurich and Swiss Federal Laboratories for Materials Science and Technology (EMPA) are thanked for access to their facilities. We thank P. Sanz Berman. for contributions to Supplementary movie 1.

## Author contributions

V.G. and J.P.-R. conceived and conceptualized the stages of this study. V.G. synthesized the catalysts, contributed to their characterization, and conducted the catalytic tests. M.A., S.K., and G.J. conducted the EPR spectroscopy studies. J.M.G.-A, and A.R.-F. performed the DFT simulations. F.K. carried out the electron microscopy investigations. F.B. performed the soft XAS analyses. Y.-T.C. and M.V. conducted the ultraviolet-visible spectroscopy and infrared spectroscopy analyses, respectively. J.P.-R., N.L., and G.J. acquired resources and funding. J.P.-R. supervised the entire project. The manuscript was written through the contributions of all authors. All authors have approved the final version of the manuscript.

## Competing interests

The authors declare no competing interests.
