## [Transparent peer review file · Nature Communications]

Tracking life and death of carbon nitride supports in platinum-catalyzed vinyl chloride synthesis

Corresponding Author: Professor Javier Pérez-Ramírez

Version 0:

Reviewer comments:

Reviewer #1

(Remarks to the Author)

This manuscript presents a comprehensive study on Pt single-atom catalysts for acetylene hydrochlorination with detailed operando characterization. The results are original and significant in the development of improved Pt catalysts for acetylene hydrochlorination, however, there are some major concerns should be addressed before I can recommend for the publication on Nat. Commun.

1. The authors failed to provide enough evidence to prove Pt species is atomically dispersed on the surface of the catalysts. The HAADF-STEM images are not enough. In fact, there are some clusters present in Figure 1b, the authors should provide the quantification analysis of different Pt species and discuss their contributions to the reactivity.
2. Figure 6. The Operando CW-EPR spectra was carried out at 298 K, while the catalytic evaluation was conducted at 473 K. The EPR results tested at 298 K are hardly representative of the actual catalytic process at 473 K. Please add the Operando CW-EPR at 473 K.
3. This work seems to conflict with their previous works (Nat Commun 12, 4016, 2021). The authors previously reported that the active metal can interact with C₂H₂, but this study stated that pyridinic N-atoms act as C₂H₂ adsorption sites. Please explain. It should be compared for the interactions between C₂H₂ and metallic site/supports.
4. The authors seem to ignore the role of Pt throughout the manuscript. The authors only mentioned the role of carbon carriers in the manuscript, while the catalytic effect of Pt as the active centre is not discussed. How Pt single atoms evolves during the reaction. How does it related with activity? How does carbon affect the stability of the oxidation state of Pt single atoms?
5. C₂H₂ is generally recognised as a precursor to coke, and the author indicated that C₂H₂ is adsorbed on carbon nitrides. Does the carbon alone undergo coke formation? What is the selectivity for these reactions? Is Pt required for the coke formation? Is there a correlation between C₂H₂ adsorption and coke build-up?

Reviewer #2

(Remarks to the Author)

In this manuscript, combining the operando EPR with other spectroscopic and kinetic analyses, the life and death of carbon nitride supports for Pt single atoms in acetylene hydrochlorination has been discussed. This work can help tracking active functionalities in carbons in acetylene hydrochlorination, but there are some results that need to be clarified. Before considering accepting the work published in Nature Communications, the authors need to issue the following questions:

1. Although several spectroscopic analyses were applied to investigate the structure of the carbon nitride supports in Figure 2, some subtle differences are still hard to distinguish, especially UV-vis DRS and Raman spectra. Thus, the soft X-ray absorption spectroscopy of N is necessary.
2. In Figure 7a, during the formation of NH₃, Does the coordination environment of Pt change in this case?
3. Line 300. The author indicated that "a NH₃ molecule that can be released into the gas phase". Add relevant experimental evidences.
4. Several work have showed obvious interactions between active metal and C₂H₂ on the Au-, Ru- and Pt-based catalysts, which seem to conflict with this work. Does Pt also interact with C₂H₂?
5. To further explore the adsorption sites for C₂H₂, contrast experiments need to be performed. Control experiments should also be carried out on different carbon carriers without added Pt. In addition, C₂H₂-TPD characterizations also need to be conducted.

Reviewer #3

(Remarks to the Author)

The study by Dr. Pérez-Ramírez et al. investigated the deactivation mechanisms of carbon nitride-supported Pt single-atom catalysts during vinyl chloride synthesis via acetylene hydrochlorination. Utilizing operando electron paramagnetic resonance spectroscopy and simulations, the study concluded that changes in the carbon nitride support due to its reaction with HCl lead to catalyst deactivation. Specifically, HCl induces carbon nitride depolymerization by protonating graphitic nitrogen atoms and generating graphitic nitrogen vacancies. These changes reduce C₂H₂ adsorption and promote radical polymerization into coke, both of which are directly responsible for the deactivation.

I think the manuscript is of good quality. It addresses an important area that was previously underexplored, and the arguments are generally well-supported. I believe the publication of this work would be of significant interest to the broader catalysis and chemistry community. However, there are several points I would like the authors to clarify before I can recommend the manuscript for publication.

1. In this study, the Pt loadings are fixed at 1 wt% for different carbon nitride samples. However, these supports have very different BET surface areas, ranging from 4 m²/g for LMO to 329 m²/g for ECN. When Pt is loaded onto these supports, the activities of samples with higher surface area supports are higher, which are attributed in part to their higher polymerization degree (fewer NH_x terminations and N₂C vacancies). However, to establish the effect of surface features associated with different polymerization degrees on the reaction, it is necessary to ensure consistent surface areas across all samples. Additionally, the potential influence of Pt surface density, which varies significantly between these samples, must be ruled out. Therefore, I recommend adding a series of experiments using samples with a constant Pt surface density and controlling the sample amount for each test to ensure that the Pt amount remains the same.

- 2.
- a) An important observation in the study is the correlation between deactivation rates and the average g-value of the paramagnetic centers in the carbon nitride supports. However, the study also shows that the g-value changes during the reaction: the key to the validity of the correlation is that the relationship still holds when g-values from measurements during the reaction are used. Therefore, instead of using g-values from fresh samples, g-values from post-catalysis samples or HCl-activated samples should be used instead.
 - b) In addition, the KD values are determined using deactivation profiles with very different starting yields in Figure 4. The fitting may not be equally reliable in each case. Therefore, I recommend controlling the starting yield to be at a comparable level when determining the KD values by adjusting the sample amount.
 - c) What is the deactivation profile beyond 24 hours? Will the deactivation continue, or will the activity gradually stabilize?

3. In Supplementary Figure 3b, both the yellow and red regions are labeled as Cl-Pt. Is that a typo?

Version 1:

Reviewer comments:

Reviewer #1

(Remarks to the Author)

The revised manuscript can be accepted as it is.

Reviewer #2

(Remarks to the Author)

The manuscript has been carefully revised, and it can be accepted under the current status.

Reviewer #3

(Remarks to the Author)

Thank you for the revision. With the new data, I believe the relationship between the presence of -NH_x terminations and N₂C-vacancies in the carbon nitride-supported platinum catalysts and their deactivation rates in vinyl chloride synthesis is now clearer.

However, regarding my first question: In Supplementary Figure 11, it appears that PtSA/ELMO shows higher initial activity than the sample with a higher degree of polymerization (PtSA/ECN). Isn't this contradictory to the statement on page 9, lines 1-7: "In line with the fewer -NH_x terminations and N₂C-vacancies, the ECN-supported Pt SAC shows higher activity, followed by the EppCN- and ELMO-supported analogs"?

Moreover, there appears to be a discrepancy between Supplementary Figure 11 and 15 regarding the relative initial activity of the three samples. Could you please confirm whether the sample labeling in Supplementary Figure 11 is correct?

Manuscript NCOMMS-24-78202-T - Response to Reviewers

Comments in blue | Replies in black | Actions in bold

Important note: Indicated page, line, or figure numbers refer to the revised manuscript and/or supplementary information with changes highlighted.

Reviewer #1

This manuscript presents a comprehensive study on Pt single-atom catalysts for acetylene hydrochlorination with detailed *operando* characterization. The results are original and significant in the development of improved Pt catalysts for acetylene hydrochlorination, however, there are some major concerns should be addressed before I can recommend for the publication on Nat. Commun.

We thank the Reviewer for acknowledging the detailed use of *operando* electron paramagnetic resonance (EPR) spectroscopy in our study and recognizing the originality and significance of the results. By addressing their constructive criticism, we were able to further strengthen the quality, clarity, and impact of our study. By conducting additional analyses by microscopy with a deep-learning based atom-detection tool and EPR spectroscopy together with catalytic testing, we were able to further clarify aspects related to the reaction mechanism and the respective contributions of the Pt species (as single atoms or nanoparticles) and carbon nitride components. Each point is addressed below with a description of the actions taken upon revision.

1. The authors failed to provide enough evidence to prove Pt species is atomically dispersed on the surface of the catalysts. The HAADF-STEM images are not enough. In fact, there are some clusters present in Figure 1b, the authors should provide the quantification analysis of different Pt species and discuss their contributions to the reactivity.

We thank the Reviewer for the comment. The predominant presence of Pt single atoms is visualized by HAADF-STEM measurements, corroborated by the absence of metallic diffraction peaks in X-ray diffractograms (page S22, Supplementary Figure 1), and further confirmed by the lack of Pt-Pt scattering contributions to the extended X-ray absorption fine structure (EXAFS) analysis of Pt_{SA}/ECN (page 33, Figure 5; page S15, Supplementary Table 7). **To further assess the metal nanostructure distribution quantitatively and without arbitrariness, we have analyzed at least 13 HAADF-STEM images collected for each catalyst by a deep-learning based atom-detection tool** (<https://huggingface.co/spaces/nccr-catalysis/atom-detection>, *Adv. Mater.* **36**, 2307991 (2024), page S24, Supplementary Figure 2; page 4, lines 20-24; page 5, lines 1-4), which determines the distribution of nearest-neighbor distances (NND) based both on advanced supervised and unsupervised learning methods. This approach leverages convolutional neural networks (CNNs) for pixel-wise metal center identification, combined with Gaussian mixture models (GMMs) to resolve overlapping features and identify low-nuclearity clusters. Though sporadic clusters are detected, the mean NND values for all catalysts exceed 0.32 nm, indicating distances much larger than the dimer value, 0.24 nm, with Pt_{SA}/ECN showing a value >0.5 nm.

To better assess their contribution to catalyzing acetylene hydrochlorination to vinyl chloride, we have synthesized Pt clusters on the exfoliated carbon nitride support, Pt_{NP}/ECN, as corroborated by HAADF-STEM analysis, and tested the catalytic performance (page S26, Supplementary Figure 4; page 5, lines 8-10). In agreement with previous studies comparing Pt single atoms and nanoparticles (*Nat. Catal.* **3**, 376 (2020); *Nat. Nanotechnol.* **17**, 606 (2022)), the of Pt_{NP}/ECN show three-fold lower catalytic activity than Pt_{tSA}/ECN. This suggests that sporadic Pt aggregates in Pt_{tSA}/ECN do not contribute significantly to fulfilling the catalytic cycle.

2. Figure 6. The Operando CW-EPR spectra was carried out at 298 K, while the catalytic evaluation was conducted at 473 K. The EPR results tested at 298 K are hardly representative of the actual catalytic process at 473 K. Please add the Operando CW-EPR at 473 K.

We thank the Reviewer for their comment, we realized that the caption of Figure 6 was not sufficiently detailed, which may have led to some confusion about the *operando* and *in situ* experimental setups. The *operando* CW-EPR measurements (page 35, Figure 6a) were conducted at 473 K, matching the catalytic evaluation temperature to ensure representative data. Conversely, the *in situ* measurements (page 35, Figure 6d) were intentionally conducted at distinct conditions. This approach was taken to disentangle the effects of temperature and reactants on the support. Specifically, when heating the catalyst to 473 K under argon, we already observed some restructuring of the carbon nitride support. Hence, to exclusively assess the influence of individual reactants, we tested the catalyst at 298 K while exposing it to HCl and C₂H₂ independently. This allowed us to determine that restructuring occurred only under HCl exposure and not under C₂H₂, avoiding the convolution of temperature and reactant effects. **We have amended the caption of Figure 6 in the revised manuscript to describe the conditions for both *operando* and *in situ* experiments (page 35, lines 3 and 6), and clarified their description in the manuscript (page 13, line 24).**

3. This work seems to conflict with their previous works (*Nat Commun* **12**, 4016, 2021). The authors previously reported that the active metal can interact with C₂H₂, but this study stated that pyridinic N-atoms act as C₂H₂ adsorption sites. Please explain. It should be compared for the interactions between C₂H₂ and metallic site/supports.

This is a central point, which also relates to comment #4 of Reviewer 1 and comment #4 of Reviewer 2. The previous work mentioned by Reviewer 1 (*Nat. Commun.* **12**, 4016 (2021)) concluded the key role of carbon in generating performing metal-based catalysts, correlating (i) the catalytic performance of carbon-supported Pt SACs with the respective chemisorption capacity and (ii) the acetylene chemisorption capacity with support compositional characteristics such as abundance and acidity of functional groups. Carbon was identified as an acetylene reservoir based on these results, while a speculative mechanism was proposed where acetylene conversion into vinyl chloride would occur over the platinum atoms. Still, owing to the highly corrosive nature of the reaction environment, requiring careful experiment design and safety assessment, neither *in situ* nor *operando* studies had been conducted – which limited the identification of the active site and their dynamic behavior. Recently, we could develop a tailored approach for *operando* X-ray

absorption spectroscopy analysis, investigating the dynamics of several Pt SACs supported on different carbon supports, including N-doped, activated and non-activated carbons (*Nat. Commun.* **14**, 5557 (2023); *ACS Catal.* **14**, 13652 (2024)). Only metal-HCl interactions could be observed while no metal-C₂H₂ bonds were ever detected. Conversely, kinetic investigations, *ex situ* X-ray photoelectron spectroscopy (XPS) analyses of the carbon support, and C₂H₂ temperature programmed desorption measurements indicated carbon-C₂H₂ interactions. Density functional theory (DFT) simulations showed that the high chlorination degree of the working PtCl_x (x = 2,3), as determined by *operando* XAS, hinders C₂H₂ adsorption over the metal sites, preferring adsorption and activation over functionalities in the carbon in proximity to the metal. **These aspects are further discussed in the Introduction section of the manuscript (page 2, lines 7-15).**

Please refer to our reply to comment #4 regarding the discussion of the role of Pt single atoms in fulfilling the catalytic cycle of acetylene hydrochlorination.

4. The authors seem to ignore the role of Pt throughout the manuscript. The authors only mentioned the role of carbon carriers in the manuscript, while the catalytic effect of Pt as the active centre is not discussed. How Pt single atoms evolves during the reaction. How does it related with activity? How does carbon affect the stability of the oxidation state of Pt single atoms?

We thank the Reviewer for raising this relevant point, which relates to comment #3. Pt single atoms are a necessary component of the catalyst, as bare carbon nitride supports are virtually inactive (page S9, Supplementary Table 1). As detailed in our reply to comment #3, this has been recently attributed to the bifunctional catalytic role of Pt atoms and carbon supports, which activate HCl and C₂H₂, respectively (*Nat. Commun.* **14**, 5557 (2023); *ACS Catal.* **14**, 13652 (2024)). Analysis by XAS of Pt_{SA}/ECN supports this (page 10, lines 21-26; page 11, lines 1-12; page 33, Figure 5; S15, Supplementary Table 7). The as-prepared Pt atoms, derived from H₂PtCl₆, exhibit high chlorination (Pt-Cl coordination number CN = 3.2) and a two-fold coordination with the carbon nitride support (Pt-N/C CN = 2.1). After use in acetylene hydrochlorination, Pt L₃ XAS measurements detect a slight increase in chlorination degree, reflecting HCl activation, and a lower Pt-N/C contribution, which is assigned to lack of Pt-C₂H₂ and a weaker one-fold coordination interaction with the support (after 3 h on stream, Pt-Cl CN = 3.4 and Pt-N/C CN = 0.8). These results agree with previous findings by *operando* XAS, indicating adaptive PtCl_x (x = 2-3) as active species that can activate HCl for VCM synthesis (*Nat. Commun.* **14**, 5557 (2023); *ACS Catal.* **14**, 13652 (2024)). Only once Pt_{SA}/ECN fully deactivated after extended use, *i.e.*, 24 h, XAS analysis shows slightly lower chlorination (CN = 2.6), indicating reduced ability to activate HCl, and marginally increased Pt-N/C (CN = 1.1) suggesting that Pt atoms might eventually suffer from minor blockage by coke deposits formed over N-vacancies in the support.

To gain further understanding of the Pt-reactant interactions during reaction, **we have conducted density functional theory analyses** probing the competitive adsorption of HCl and C₂H₂ over the pristine Pt atom architecture, featuring two chloride ligands and a two-fold coordination with the support, and working Pt sites, which are bichlorinated and coordinated one- or two-fold with the carbon nitride support presenting N_{3C}-vacancies. Simulations show that C₂H₂ adsorption is

hindered in the presence of HCl (by up to 0.26 eV) or even fully inhibited by chloride ligands (resulting in weak physisorption with Pt–C₂H₂ distance exceeding 3 Å). **These results have been included in the revised manuscript and supplementary information (page 15, lines 7-13; page S51, Supplementary Figure 29).**

Moreover, to probe the changes in the oxidation state of working Pt sites, **we have computed the Bader charges** of pristine PtCl₂ and upon HCl adsorption, considering one- and two-fold coordination with the carbon nitride support. These calculations show that there is no major alteration in the electronic properties of Pt upon HCl activation (<0.18 |e⁻|), in line with the analyses by XPS of the oxidation state in Pt_{SA}/ECN as-prepared and after use in acetylene hydrochlorination for 24 h (page 15, lines 14-16; page S25, Supplementary Figure 3; page S11, Supplementary Table 3). This highlights the role of the carbon nitride in controlling the electronic properties of the Pt atoms, even upon HCl activation, by virtue of its conductive nature allowing for metal-support electron density redistribution. **These findings have been discussed in the revised manuscript and supplementary information (page 15, lines 13-18; page S18, Supplementary Table 9).**

5. C₂H₂ is generally recognised as a precursor to coke, and the author indicated that C₂H₂ is adsorbed on carbon nitrides. Does the carbon alone undergo coke formation? What is the selectivity for these reactions? Is Pt required for the coke formation? Is there a correlation between C₂H₂ adsorption and coke build-up?

We thank the Reviewer for the pertinent questions, which also relate to comment #5 of Reviewer #2. As stated in the supplementary information (page S6, lines 9-10), the error of the carbon balance between the acetylene at the reactor inlet and the acetylene and vinyl chloride at the reactor outlet was less than 10% in all experiments, *i.e.*, the carbon mass balance was closed at ≥90%. The minor conversion of the acetylene at the reactor inlet into coke deposits, which are not accounted for in the carbon balance, is corroborated by thermogravimetric analysis (TGA).

Bare carbon nitride supports exhibit very low catalytic activity in acetylene hydrochlorination (page S9, Supplementary Table 1), attributable to the lack of metal sites that activate HCl. Still, to gain deeper insights into deactivation pathways in bare ECN, **we have tested its catalytic performance over 24 h and analyzed the used catalyst by electron paramagnetic spectroscopy (page S49, Supplementary Figure 27; page S50, Supplementary Figure 28; page 13, lines 18-23).** As elaborated in our reply to comment #5 of Reviewer #2, we observe catalyst deactivation that can be assigned to metal-independent restructuring – as coking and depolymerization – of the bare carbon nitride support.

These results further highlight the critical role of the carbon support in Pt SACs. Hence, **we have conducted TGA measurements of exfoliated carbon nitride supported Pt SACs, as-prepared and after use in acetylene hydrochlorination for 24 h (page S35, Supplementary Figure 13).** The coke amount formed during reaction correlates with the C₂H₂ adsorption capacity of the bare carbon nitride supports, linking the latter one with deactivation of the corresponding Pt SAC deactivation. **These results are now included in the revised manuscript and supplementary information (page 9, lines 18-19; page S35, Supplementary Figure 13).**

Reviewer #2

In this manuscript, combining the operando EPR with other spectroscopic and kinetic analyses, the life and death of carbon nitride supports for Pt single atoms in acetylene hydrochlorination has been discussed. This work can help tracking active functionalities in carbons in acetylene hydrochlorination, but some results need to be clarified. Before considering accepting the work published in Nature Communications, the authors need to issue the following questions:

We thank the Reviewer for acknowledging that the approach developed in our study advances tracking active sites in carbons in acetylene hydrochlorination. Their thoughtful comments, addressed below, prompted us to conduct additional kinetic, computational, and spectroscopic analyses to further elaborate on the dynamics and catalytic role of both Pt single atoms and functionalities in the carbon nitride support.

1. Although several spectroscopic analyses were applied to investigate the structure of the carbon nitride supports in Figure 2, some subtle differences are still hard to distinguish, especially UV-vis DRS and Raman spectra. Thus, the soft X-ray absorption spectroscopy of N is necessary.

We thank the Reviewer for the suggestion. While spectral features across the differently polymerized carbon nitrides (*i.e.*, Pt_{SA}/CN and Pt_{SA}/LMO) are clear, those of their exfoliated counterparts are more subtle (page 5, lines 18-26; page 6, lines 1-9). Therefore, **we have conducted soft X-ray absorption spectroscopy (XAS) analysis of Pt_{SA}/ECN and Pt_{SA}/ELMO at the nitrogen K edge** to further investigate the two carbon nitride matrices (page 30, Figure 2d; page 6, lines 9-24; page S3, lines 35-36; page S4, lines 1-2). Both supports consist of polymeric units derived from heptazine rings, with varying degree of –NH_x terminations and N-vacancies. In line with this and the bulk-averaging nature of XAS, the spectra of Pt_{SA}/ECN and Pt_{SA}/ELMO exhibit similar features. In agreement with literature reports (*Small Methods* 5 2000707 (2021)), we note three spectral contributions to the X-ray absorption near edge structure (XANES): at 399.6 eV (N1), 401.5 eV (N2), and 402.3 eV (N3). N1 is assigned to the N 1s → π* transition in aromatic N_{2C}-atoms of heterocyclic rings, π* (C=N–C); N2 to graphitic N_{3C}-atoms, π*(N–3C); and N3 to sp³ (potentially protonated) N_{3C,link}-atoms, π*(N–C), respectively. Still, the shape of the N1 contribution appears to be broader, shifting to higher energies, in Pt_{SA}/ELMO than Pt_{SA}/ECN. By comparison with reference materials featuring protonated N-atoms, dicyandiamide and melamine, which exhibit spectral contributions at 399.7 and 399.9 eV, respectively, the larger shoulder of the N1 contribution in Pt_{SA}/ELMO is tentatively attributed to higher protonation of the support, *i.e.*, more –NH_x terminations.

2. In Figure 7a, during the formation of NH₃, Does the coordination environment of Pt change in this case?

We thank the Reviewer for the relevant question, which also relates to comment #4 of Reviewer #1. The formation of NH₃ occurs at the support, which can be facilitated by the activation of HCl over Pt atoms with subsequent transfer of H onto the carbon nitride matrix. As detailed in the manuscript (page 10, lines 21-26; page 11, lines 1-8), XAS analysis resolves the structure of pristine

metal sites as chlorinated Pt atoms coordinated two-fold with the carbon nitride support. Upon reaction, we observe slight chlorination, reflecting HCl activation, and a lower Pt-N/C contribution, attributed to resulting weaker one-fold coordination interaction with the support and indicating lack of Pt–C₂H₂ interactions. Hence, **we have conducted additional DFT simulations** to investigate the formation of N_{3C}-vacancies in the presence of chlorinated Pt atoms both one- and two-fold coordinated with the support, as both species can co-exist during reaction. Our simulations yield comparable energy profiles, **which have been included in the revised manuscript (page 15, lines 21-26; page 16, lines 1-11; page 37, Figure 7a)**, underscoring the central role of N-functionalities in the carbon nitride in driving the matrix restructuring *via* protonation.

3. Line 300. The author indicated that “a NH₃ molecule that can be released into the gas phase”. Add relevant experimental evidences.

We thank the Reviewer for the valuable comment. **We have conducted online mass spectrometry analyses to gain insights into time-resolved product formation, which have been included in the revised manuscript and supplementary information (page 10, lines 4-12; page 32, Figure 4b; page S38, Supplementary Figure 16)**. Both the C₂H₂ and HCl signals gradually increase over time on stream, while the VCM signal rises, indicating catalyst deactivation. Still, the C₂H₂ signal initially plateaus (page S38, Supplementary Figure 16), suggesting that C₂H₂ is consumed both in VCM and coke formation. In agreement with simulations (page 15, line 25; page 16, lines 1-8), another product, NH₃, is detected. Its formation gradually decreases over time, with kinetics that appear to align with HCl consumption. This further corroborates that NH₃ formation results from protonation of the N-functionalities in the carbon nitride support, as rationalized by DFT simulations (page 37, Figure 7a).

4. Several work have showed obvious interactions between active metal and C₂H₂ on the Au-, Ru- and Pt-based catalysts, which seem to conflict with this work. Does Pt also interact with C₂H₂?

We thank the Reviewer for the pertinent comment. Please refer to our replies to comments #3 and #4 of Reviewer #1.

5. To further explore the adsorption sites for C₂H₂, contrast experiments need to be performed. Control experiments should also be carried out on different carbon carriers without added Pt. In addition, C₂H₂-TPD characterizations also need to be conducted.

We appreciate the valuable suggestions of the Reviewer, which also relate to comment #5 of Reviewer #1. We clarify that catalytic testing of the bare carbon nitride supports, without Pt, were already conducted as shown in Supplementary Table 1 and discussed in the manuscript (page 9, lines 12-14). Owing to the lack of Pt sites that can facilitate HCl activation, the bare carbon nitride supports prove virtually inactive for VCM synthesis. To complement our analysis of the C₂H₂ adsorption properties of the carbon nitrides by chemisorption measurements (page 8, lines 20-23; page 31, Figure 3), **we have conducted temperature-programmed desorption analyses of C₂H₂ with mass spectrometry (C₂H₂-TPD-MS; page S43, Supplementary Figure 21; page 12, lines 3-12)**. While C₂H₂ desorption is observed from 350 to 500 K, the carbon nitride support starts

undergoing thermal decomposition from 500 K onward, hindering the detection of desorbed C₂H₂. In fact, the C₂H₂ MS signal (26 m/z) overlaps with that of cyanide, which originates from the decomposition of ECN, as evidenced by the peak detected when analyzing the carbon nitride support flowing He only, without C₂H₂. These results further corroborate the proneness of carbon nitride to restructuring, and indicate that C₂H₂-TPD-MS does not constitute a suitable method for analyzing C₂H₂ adsorption over carbon nitrides.

Still, to gain deeper insights into the interaction of carbon nitrides with C₂H₂ by contrast experiments, **we have performed *ex situ* EPR measurements of the bare ECN support as-prepared and after use in acetylene hydrochlorination for 24 h and compared the obtained spectra with the Pt_{SA}/ECN counterparts (page S49, Supplementary Figure 27; page S50, Supplementary Figure 28; page 13, lines 18-23).** A reaction-induced low-field shift of the CW-EPR signal for the bare ECN support is observed, analogous to the one previously detected for Pt_{SA}/ECN catalyst (page 33, Figure 5). This indicates that the carbon nitride support can interact with the reactants, and restructure, independently of the metal sites, which further underscores the key role of the carbon support in regulating the performance of SACs in acetylene hydrochlorination.

Reviewer #3

The study by Dr. Pérez-Ramírez et al. investigated the deactivation mechanisms of carbon nitride-supported Pt single-atom catalysts during vinyl chloride synthesis via acetylene hydrochlorination. Utilizing operando electron paramagnetic resonance spectroscopy and simulations, the study concluded that changes in the carbon nitride support due to its reaction with HCl lead to catalyst deactivation. Specifically, HCl induces carbon nitride depolymerization by protonating graphitic nitrogen atoms and generating graphitic nitrogen vacancies. These changes reduce C_2H_2 adsorption and promote radical polymerization into coke, both of which are directly responsible for the deactivation. I think the manuscript is of good quality. It addresses an important area that was previously underexplored, and the arguments are generally well-supported. I believe the publication of this work would be of significant interest to the broader catalysis and chemistry community. However, there are several points I would like the authors to clarify before I can recommend the manuscript for publication.

We warmly thank the Reviewer for recognizing the quality of our study and arguments. We are pleased that our findings on the unexplored area of carbon support catalytic activity are deemed valuable to the catalysis and chemistry community. The Reviewer's constructive comments have been highly valuable in enhancing the clarity and impact of our manuscript. Below, we address each inquiry, along with the corresponding actions taken.

1. In this study, the Pt loadings are fixed at 1 wt% for different carbon nitride samples. However, these supports have very different BET surface areas, ranging from 4 m^2/g for LMO to 329 m^2/g for ECN. When Pt is loaded onto these supports, the activities of samples with higher surface area supports are higher, which are attributed in part to their higher polymerization degree (fewer NH_x terminations and N_2C vacancies). However, to establish the effect of surface features associated with different polymerization degrees on the reaction, it is necessary to ensure consistent surface areas across all samples. Additionally, the potential influence of Pt surface density, which varies significantly between these samples, must be ruled out. Therefore, I recommend adding a series of experiments using samples with a constant Pt surface density and controlling the sample amount for each test to ensure that the Pt amount remains the same.

We thank the Reviewer for the insightful comment. As stated in the manuscript (page 8, lines 20-23), the surface area of the carbon nitride plays a role in regulating the adsorption, and thus activation, of C_2H_2 . Regardless of their polymerization degrees, the carbon nitride matrices are found to lead to appreciable activity in Pt SACs only if thermally exfoliated (and thus exhibiting sufficient surface area for C_2H_2 adsorption). Within the series of exfoliated carbon nitriles (ELMO, EppCN, and ECN), their synthesis method *via* thermal exfoliation cannot inherently ensure constant surface area across differently polymerized matrices, since the exfoliation degree cannot be controlled independently of the polymerization degree. Thermal exfoliation of a low-polymerized matrix tends to yield lower surface area, but a longer treatment or one at higher temperature would inevitably induce simultaneous matrix polymerization. Still, while surface area certainly contributed to regulating the catalytic activity of carbon nitride-supported Pt SACs, other features

relating to the support polymerization degree also play a role. This is reflected not only in the initial activity of the Pt SACs but also in their performance on stream. The virtual inactivity of $-\text{NH}_x$ terminations is corroborated by the deactivation mechanism *via* carbon nitride depolymerization evidenced by *operando* EPR and DFT simulations (page 12, lines 13-26; page 16, lines 1-8; page 35, Figure 6; page 37, Figure 7). The detrimental impact of $\text{N}_{2\text{C}}$ -vacancies is supported by the correlation between the average g factor of carbon nitride supports (linked to more asymmetric $\text{N}_{2\text{C}}$ -vacancies and lower polymerization) and deactivation constants of the corresponding Pt SACs (page 32, Figure 4a; page 9, lines 19-22). To further investigate these aspects, following the Reviewer's valuable suggestion, **we have conducted catalytic tests over Pt SACs supported on the exfoliated carbon nitride series, keeping a constant metal-content-to-surface-area ratio (i.e., metal density, $75 \mu\text{mol}_{\text{Pt}} \text{m}^{-2}$) and varying the catalyst mass, and thus space velocity, to maintain a constant reactant flow rate per metal site (page S33, Supplementary Figure 11; page 9, lines 1-7).** In line with the fewer $-\text{NH}_x$ terminations and $\text{N}_{2\text{C}}$ -vacancies, the ECN-supported Pt SAC shows higher activity, followed by the EppCN- and ELMO-supported analogs.

2a) An important observation in the study is the correlation between deactivation rates and the average g -value of the paramagnetic centers in the carbon nitride supports. However, the study also shows that the g -value changes during the reaction: the key to the validity of the correlation is that the relationship still holds when g -values from measurements during the reaction are used. Therefore, instead of using g -values from fresh samples, g -values from post-catalysis samples or HCl-activated samples should be used instead.

We thank the Reviewer for the pertinent comment. We highlight that the g values of the as-prepared Pt SACs reflect the structural properties of the pristine catalytic surfaces, which, in turn, correlate directly with the initial deactivation rate. Still, in response to the Reviewer's suggestion, **we have included CW-EPR spectra of the Pt SACs after use in acetylene hydrochlorination over 24 h in the revised manuscript (page S36, Supplementary Figure 14; page 9, lines 21-22).** The g factors from the used catalysts exhibit a similar trend to those observed in the as-prepared samples. The more pronounced difference in g factor between $\text{Pt}_{\text{SA}}/\text{ECN}$ and $\text{Pt}_{\text{SA}}/\text{EppCN}$ or $\text{Pt}_{\text{SA}}/\text{ELMO}$ (which show similar g factors) aligns with the lesser deactivation observed for $\text{Pt}_{\text{SA}}/\text{ECN}$. The higher g factors in the used $\text{Pt}_{\text{SA}}/\text{EppCN}$ and $\text{Pt}_{\text{SA}}/\text{ELMO}$ correlate with their near-complete deactivation after 24 h on stream, further confirming that a higher presence of $-\text{NH}_x$ terminations and $\text{N}_{2\text{C}}$ -vacancies contributes to more significant catalyst deactivation.

2b) In addition, the K_{D} values are determined using deactivation profiles with very different starting yields in Figure 4. The fitting may not be equally reliable in each case. Therefore, I recommend controlling the starting yield to be at a comparable level when determining the K_{D} values by adjusting the sample amount.

We appreciate the Reviewer's valuable suggestion. **We have conducted catalytic tests varying the catalyst mass, and thus space velocity, to obtain a comparable VCM yield across the different catalysts (page S37, Supplementary Figure 15; page 9, lines 24-26).** The deactivation constants

determined by exponential regression exhibit similar values to those previously presented in Figure 4a (page 32), confirming the relationship between the polymerization degree of the carbon nitride matrix and the deactivation rate of the Pt SAC (page 32, Figure 4a). Furthermore, the catalytic tests showed that catalyst deactivation proceeds over time on stream until virtual inactivity is achieved, suggesting complete fouling of the carbon nitride supports by reaction-induced depolymerization and coking (page 13, lines 10-17).

2c) What is the deactivation profile beyond 24 hours? Will the deactivation continue, or will the activity gradually stabilize?

Please refer to our reply to comment #2b.

3. In Supplementary Figure 3b, both the yellow and red regions are labeled as CI-Pt. Is that a typo? We thank the Reviewer for pointing out the typo, which has been corrected (page S25, Supplementary Figure 3).

Manuscript NCOMMS-24-78202A- Response to Reviewers

Comments in *blue* - Replies in black - Actions in **bold**

Important note: Indicated page, line, or figure numbers refer to the revised manuscript and/or Supplementary Information with changes highlighted.

Reviewer #1

The revised manuscript can be accepted as it is.

We thank Reviewer #1 for their positive feedback on our revised contribution.

Reviewer #2

The manuscript has been carefully revised, and it can be accepted under the current status.

We thank Reviewer #2 for recognizing our efforts to address the constructive comments of our peers and strengthen the quality of our contribution.

Reviewer #3

Thank you for the revision. With the new data, I believe the relationship between the presence of -NH_x terminations and N₂C-vacancies in the carbon nitride-supported platinum catalysts and their deactivation rates in vinyl chloride synthesis is now clearer.

We appreciate Reviewer #3's thoughtful assessment and recognition of the revisions made to improve the clarity of our work.

However, regarding my first question: In Supplementary Figure 11, it appears that Pt_{SA}/ELMO shows higher initial activity than the sample with a higher degree of polymerization (Pt_{SA}/ECN). Isn't this contradictory to the statement on page 9, lines 1-7: "In line with the fewer -NH_x terminations and N₂C-vacancies, the ECN-supported Pt SAC shows higher activity, followed by the EppCN- and ELMO-supported analogs"? Moreover, there appears to be a discrepancy between Supplementary Figure 11 and 15 regarding the relative initial activity of the three samples. Could you please confirm whether the sample labeling in Supplementary Figure 11 is correct?

We thank the Reviewer for pointing out the labeling mistake in Supplementary Figure 11, **which has been corrected (page S30, Supplementary Figure 11)**. The activity trend across Pt_{SA}/ECN, Pt_{SA}/EppCN, and Pt_{SA}/ELMO shown in Supplementary Figure 11 is, in fact, consistent with that in Supplementary Figure 15.